# Multi-tissue network analysis reveals the effect of JNK inhibition on dietary sucrose-induced metabolic dysfunction in rats

Hong Yang[1], Cheng Zhang[1], Woonghee Kim[1], Mengnan Shi[1], Metin Kiliclioglu[2], Cemil Bayram[3], Ismail Bolar[2], Özlem Özdemir Tozlu[4], Cem Baba[4], Nursena Yuksel[4], Serkan Yildirim[2,5], Shazia Iqbal[6], Jihad Sebhaoui[6], Ahmet Hacımuftuoglu[3], Matthias Uhlen[1], Jan Boren[7], Hasan Turkez[8], Adil Mardinoglu[1,9]*

[1]Science for Life Laboratory, KTH Royal Institute of Technology, Stockholm, Sweden; [2]Department of Pathology, Veterinary Faculty, Atatürk University, Erzurum, Turkiye; [3]Department of Medical Pharmacology, Faculty of Medicine, Atatürk University, Erzurum, Turkiye; [4]Department of Molecular Biology and Genetics, Faculty of Science, Erzurum Technical University, Erzurum, Turkiye; [5]Department of Pathology, Faculty of Veterinary Medicine, Kyrgyz-Türkish Manas Üniversity, Bishkek, Kyrgyzstan; [6]Trustlife Labs Drug Research and Development Center, Istanbul, Turkiye; [7]Department of Molecular and Clinical Medicine, University of Gothenburg, Sahlgrenska University Hospital, Gothenburg, Sweden; [8]Department of Medical Biology, Faculty of Medicine, Atatürk University, Erzurum, Turkiye; [9]Centre for Host-Microbiome Interactions, Faculty of Dentistry, Oral & Craniofacial Sciences, King's College London, London, United Kingdom

*For correspondence:
adilm@kth.se

## eLife Assessment

The authors implement a **valuable** multi-tissue approach to dissect the physiologic consequences of JNK inhibition in parallel with dietary perturbation via sucrose. The conclusions of disrupted liver, muscle and adipose metabolism being central to these effects are **solid**, as they are supported by a combination of experimental dissection and network modeling approaches.

**Abstract** Excessive consumption of sucrose, in the form of sugar-sweetened beverages, has been implicated in the pathogenesis of metabolic dysfunction-associated fatty liver disease (MAFLD) and other related metabolic syndromes. The c-Jun N-terminal kinase (JNK) pathway plays a crucial role in response to dietary stressors, and it was demonstrated that the inhibition of the JNK pathway could potentially be used in the treatment of MAFLD. However, the intricate mechanisms underlying these interventions remain incompletely understood given their multifaceted effects across multiple tissues. In this study, we challenged rats with sucrose-sweetened water and investigated the potential effects of JNK inhibition by employing network analysis based on the transcriptome profiling obtained from hepatic and extrahepatic tissues, including visceral white adipose tissue, skeletal muscle, and brain. Our data demonstrate that JNK inhibition by JNK-IN-5A effectively reduces the circulating triglyceride accumulation and inflammation in rats subjected to sucrose consumption. Coexpression analysis and genome-scale metabolic modeling reveal that sucrose overconsumption primarily induces transcriptional dysfunction related to fatty acid and oxidative metabolism in the liver and adipose tissues, which are largely rectified after JNK inhibition at a clinically relevant dose.

Skeletal muscle exhibited minimal transcriptional changes to sucrose overconsumption but underwent substantial metabolic adaptation following the JNK inhibition. Overall, our data provides novel insights into the molecular basis by which JNK inhibition exerts its metabolic effect in the metabolically active tissues. Furthermore, our findings underpin the critical role of extrahepatic metabolism in the development of diet-induced steatosis, offering valuable guidance for future studies focused on JNK-targeting for effective treatment of MAFLD.

## Introduction

Hepatic steatosis (HS) is characterized by the accumulation of triglycerides (TG) in the liver (at least 5% of liver weight) in the absence of secondary contributing factors (*Nassir et al., 2015*). MAFLD encompasses a spectrum of conditions, ranging from simple HS to metabolic dysfunction-associated steatohepatitis (MASH), which is characterized by HS combined with various degrees of liver inflammation and liver injury, with or without fibrosis. MASH can progress to advanced liver disease including cirrhosis and hepatocellular carcinoma (*Machado and Cortez-Pinto, 2023*). The global prevalence of MAFLD is estimated to be 20–25% (*Younossi et al., 2023*), and it is rapidly increasing in parallel with the epidemic of obesity and type 2 diabetes mellitus (*Kotsiliti, 2023*; *Le et al., 2023*). Given the absence of approved pharmacological therapy for MAFLD by the US Food and Drug Treatment (FDA) (*Llovet et al., 2023*; *Nassir et al., 2015*), the discovery and development of effective therapies are of great importance in MAFLD treatment.

The progression of MAFLD is associated with various factors, including overnutrition, genetic determinants, and associated comorbidities (*Sveinbjornsson et al., 2022*; *Yki-Järvinen et al., 2021*). Strong evidence from meta-analysis, prospective cohort studies, and observational studies indicate that excessive sucrose, a disaccharide that comprises equal amounts of glucose and fructose and is prevalent in all kinds of sugar-sweetened beverages, maybe an initial factor for the development of MAFLD (*Geidl-Flueck et al., 2021*; *Geurtsen et al., 2021*; *Karlsen et al., 2022*; *Malik and Hu, 2022*; *Zhang et al., 2021*). For instance, a previous study reported that rats subjected to a 12 wk high-sucrose diet (25% w/v sucrose), developed a metabolic syndrome phenotype characterized by glucose intolerance, HS, and insulin resistance (IR) (*Sousa et al., 2018*). Recent research further reveals that the consumption of sucrose at a concentration of 10% w/v can disrupt energy balance and induce HS within 10–20 wk (*Jang et al., 2020*; *Stephenson et al., 2022*). In this context, JNK, a mitogen-activated protein kinase family member, emerges as a critical player due to its known function in response to intra- and extracellular stress, including hypercaloric diet consumption (*Nikolic et al., 2020*; *Sabio et al., 2009*; *Softic et al., 2020*). Indeed, accumulating evidence implicated JNK in the pathogenesis of diet-induced obesity, IR, development of metabolic syndrome, and MAFLD (*Czaja, 2010*; *Sabio and Davis, 2010*; *Seki et al., 2012*; *Singh et al., 2009*). JNK has three isoforms: JNK1 and JNK2, which are ubiquitously expressed, and JNK3 is specifically expressed in the brain, testis, and heart (*Chang and Karin, 2001*). Previously, Singh et al. demonstrated that increased hepatic JNK signaling occurred simultaneously with steatosis and lipid accumulation in a steatohepatitis model induced by a methionine- and choline-deficient diet (MCD) (*Singh et al., 2009*). Notably, MCD diet-fed JNK1 null mice exhibited significantly reduced levels of key hallmarks of MAFLD, including HS, TG accumulation, inflammation, lipid peroxidation, hepatocyte injury, and apoptosis (*Singh et al., 2009*). Similarly, high-fat diet-fed mice with hepatic ablation of both JNK1 and JNK2 are protected from HS and IR (*Vernia et al., 2014*). Altogether, these findings suggest that the suppression of the JNK pathway may prevent the development of HS and related metabolic syndrome (*Kodama and Brenner, 2009*; *Schattenberg et al., 2006*). However, whether JNK mediates the metabolic disruption induced by sucrose overconsumption remains to be elucidated. Moreover, our previous study demonstrated that JNK-IN-5A, a selective JNK2 and JNK3 inhibitor (*Angell et al., 2007*; *Cerbone et al., 2012*), effectively decreased fat accumulation and steatosis-related protein expression in an in vitro steatosis model (*Kim et al., 2023*). However, the inhibitory effect of JNK-IN-5A to JNK and JNK-related pathways in in vivo is unknown. It is established that JNK exerts its metabolic regulation as a result of multi-tissue communication, including those between liver and adipose tissues (*Azzu et al., 2020*), and skeletal muscle (*Nikolic et al., 2020*; *Vernia et al., 2014*). A systematic investigation into tissue-specific and tissue-crosstalk metabolic changes in response to diet and therapy interventions is, therefore, paramount by using genome-scale metabolic models (GEMs). GEMs are the collection

of biochemical reactions and the associated enzymes and transporters between the cellular compartments and such models can be used in the analysis of omics data for gaining insights and interpretation of the omics data (*Mardinoglu et al., 2013*; *Mardinoglu et al., 2018*; *Mardinoglu and Nielsen, 2012*; *Mardinoglu and Palsson, 2025*).

In this study, we first performed a 7 d oral toxicology study and studied the tolerability of JNK-IN-5A in rats. Next, we explored the potential therapeutic effect of JNK-IN-5A under high sucrose consumption in in vivo by performing systems biology analysis. To elucidate the systems-wide impact of sucrose overconsumption and JNK-IN-5A treatment, we investigated the intra- and inter-tissue response through integrative analysis of transcriptomics profiles from the metabolically active tissues, including liver and three other extrahepatic tissues: visceral white adipose tissue (vWAT), skeletal muscle (SkM), and brain tissues. We further predicted the genome-wide metabolic activity changes induced by liquid sucrose ingestion and JNK-IN-5A treatment by performing genome-scale metabolic modeling.

## Results

### The acute effect of the JNK-IN-5A in rats

We previously showed that the JNK-IN-5A can effectively decrease the TG accumulation and steatosis-related protein levels in an in vitro model simulating the activated de novo lipogenesis in MAFLD patients (*Kim et al., 2023*). Here, we first performed a 7 d oral toxicology study and studied the tolerability of JNK-IN-5A in rats. Four groups of three males and three females were given vehicle and doses of 30, 100, and 300 mg/kg of JNK-IN-5A, once a day for 7 d (*Figure 1*, *Figure 2a*). We assessed clinical adverse effects, body weight, and clinical pathological parameters during the study.

A few animals including those given vehicles displayed noisy respiration and reflux. These signs were only observed in connection with dosing, suggesting being related to the high viscosity of the formulation and foul taste, and are not considered as systemic side effects of the JNK-IN-5A administration. We observed none of the administrated doses led to significant changes in body weight and hematological parameters (*Supplementary file 1*). Clinical chemistry parameters revealed lower plasma levels of glucose, blood urea nitrogen, and total proteins in female rats given JNK-IN-5A at all three doses (*Figure 2a*, *Supplementary file 1*). In addition, the two higher doses of JNK-IN-5A decreased the levels of sodium in female rats and elevated the level of phosphate in both sexes (*Figure 2a*, *Supplementary file 1*). Overall, the administration of JNK-IN-5A is well tolerated with regard to adverse clinical signs, body weight, and clinical hematology parameters.

### Treatment of liquid sucrose-induced fatty liver with JNK-IN-5A

We next investigated the effect of JNK-IN-5A treatment in a rat model of MAFLD induced by liquid sucrose ingestion (*Figure 1b*). The rats feeding a standard chow diet received either tap water (control group) or sucrose-containing water (10% w/v sucrose, sucrose group) for 4 wk, after which rats on sucrose-watering were further divided into subgroups: sucrose-only group and JNK-IN-5A-treated groups at a regular dosage of 30 mg/kg/d (sucrose +JNK_D1 group) or a higher dosage of 60 mg/kg/d (sucrose +JNK_D2 group), for 3 wk. A total of 44 rats were used in this study (n=11 rats per group).

We observed that liquid sucrose consumption resulted in significantly increased plasma levels of TG (p<0.0001) and lactate dehydrogenase (LDH, p<0.0001), which is an inflammation marker, compared to the healthy controls (*Figure 2b*, *Supplementary file 2*). On sucrose water, JNK-IN-5A-treated rats exhibited significantly lower levels of TG (JNK_D1, p<0.0001; JNK_D2, p<0.001), aspartate amino-transferase (AST; JNK_D1, p=0.09; JNK_D2, p=0.005), and LDH (JNK_D1, *n.s.*; JNK_D2, p=0.015) but had no significant change on body weight, glucose, high-density lipoprotein (HDL) and low-density lipoprotein (LDL) as compared to their corresponding controls (*Figure 2b*, *Supplementary file 2*). We also performed neurological tests to study the potential effect of JNK-IN-5A on the brain. We found no significant differences among groups in retention latencies, a measure of learning and memory abilities in passive avoidance test (*Supplementary file 3*). Additionally, the locomotor activity test was used to analyze behaviors such as locomotion, anxiety, and depression in rat. No significant differences were observed among groups in stereotypical movements, ambulatory activity, rearing, resting percentage, and distance traveled (*Supplementary file 4*). Similarly, the elevated plus maze test (*Walf*

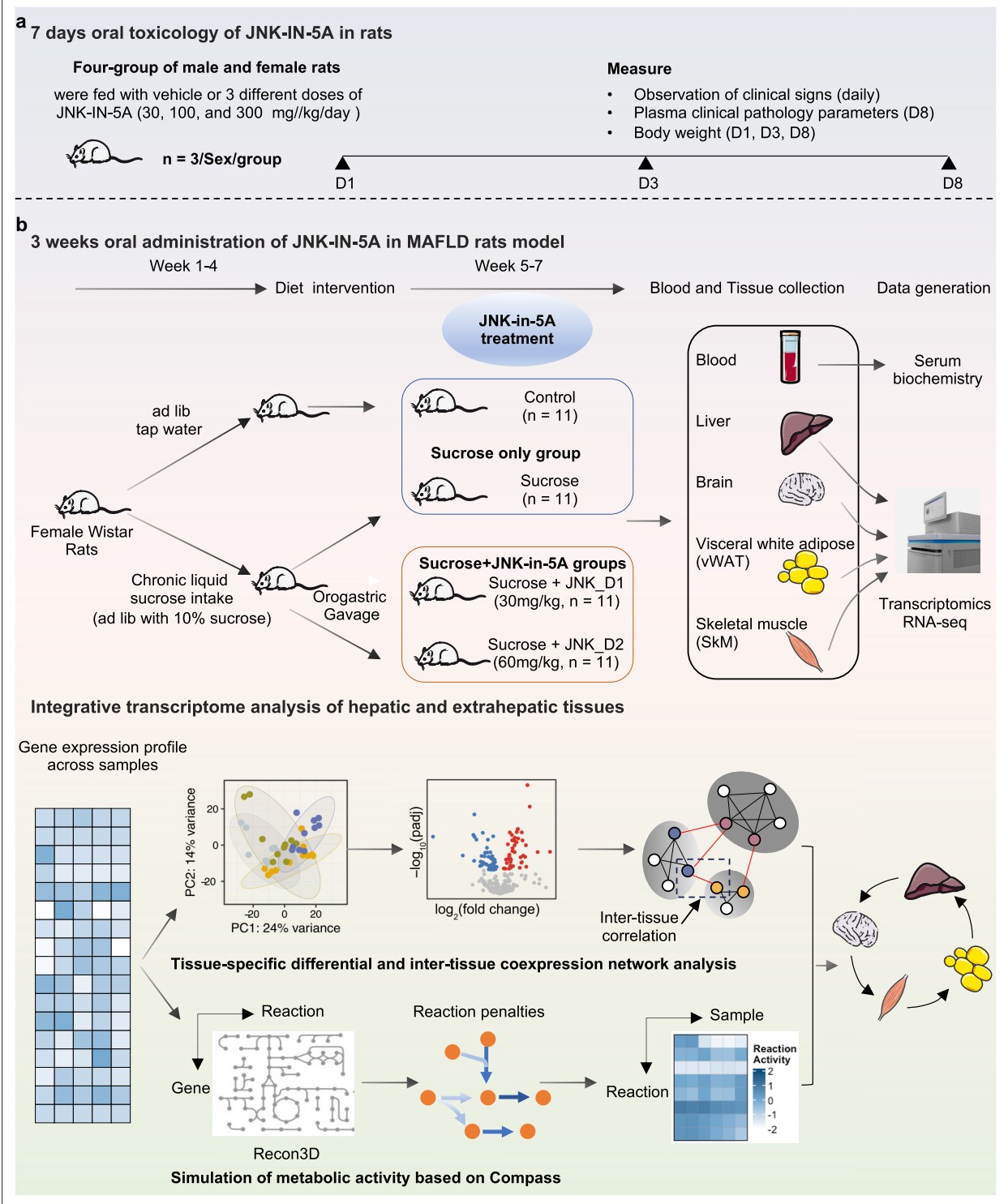

**Figure 1.** Schematic representation of experimental setup. Study groups included the healthy control group received tap water, the sucrose group received 10% sucrose water, the sucrose+JNK_D1 group received 10% sucrose and 30 mg/kg/d JNK-IN-5A, and the sucrose+JNK_D2 group received 10% sucrose and 60 mg/kg/d JNK-IN-5A (N=44, n=11 per group).

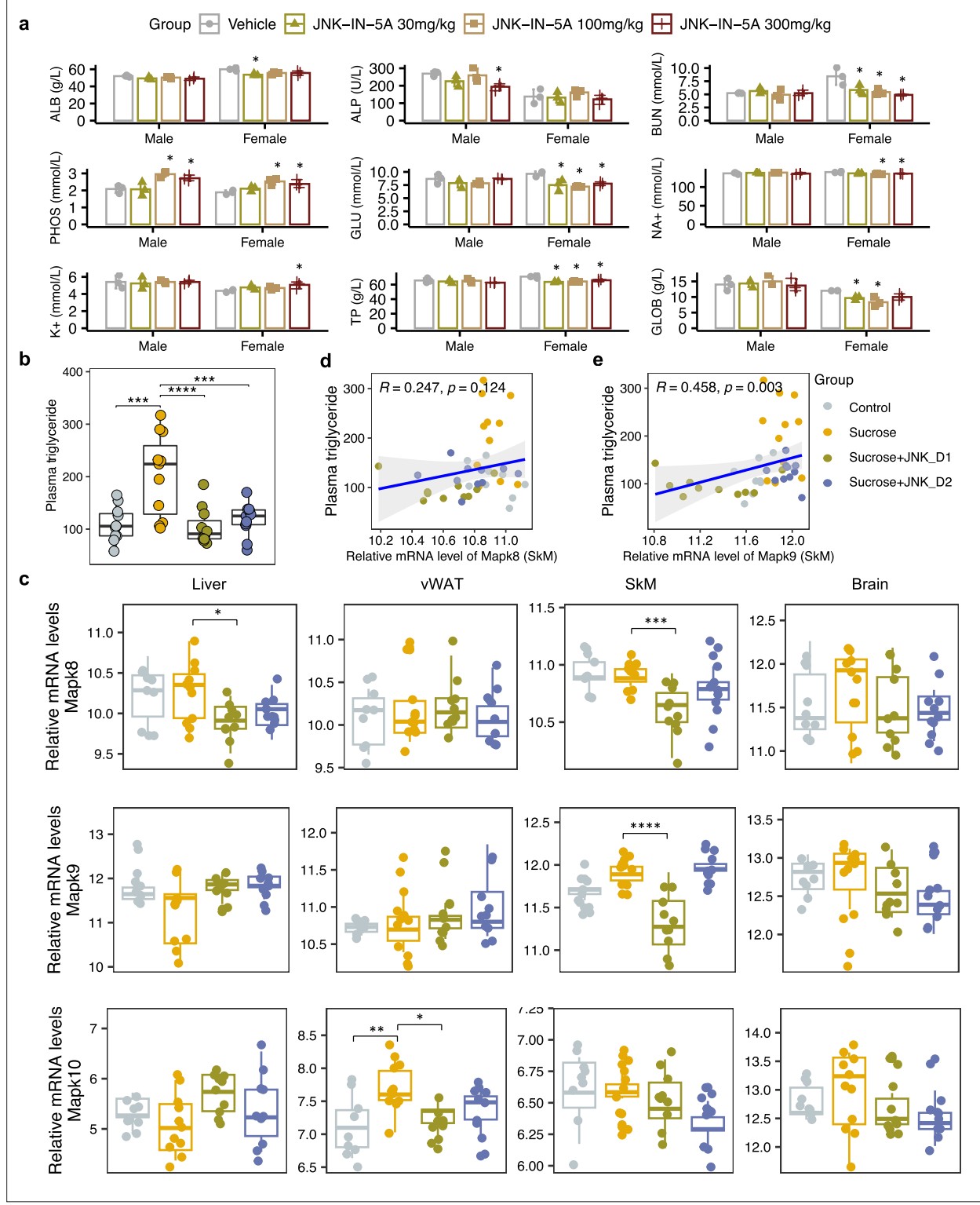

**Figure 2.** Sucrose consumption and JNK-IN-5A treatment exhibit distinct effects on the mRNA expression of genes encoding c-Jun N-terminal kinase (JNK) isoforms in the major metabolic tissues. (**a**) The acute effect of JNK-IN-5A in rats (n=3/Sex/Group). Statistically significant changed clinical chemistry parameters were presented as mean ± standard deviation (SD), see also **Supplementary file 1**. ALB: albumin; ALP: alkaline phosphatase; BUN: blood urea nitrogen; PHOS: phosphate; GLU: glucose; $K^+$: potassium; $Na^+$: sodium; TP: total Protein; GLOB: globulin. Statistical significance was assessed by one-way ANOVA testing followed by Tukey's multiple comparisons post-test or Shapiro–Wilk testing followed by Dunn's post-hoc test, as appropriate, <i>p-value <0.05 is considered as statistical significance. (**b**) Boxplot showing the levels of plasma triglycerides in control, sucrose,

*Figure 2 continued*

sucrose+JNK_D1, and sucrose +JNK_D2 rats. Statistical significance was assessed by one-way ANOVA testing followed by Tukey's multiple comparisons post-test or the Kruskal-Wallis test followed by Dunn's multiple comparisons post-test, as appropriate, *i*p-value <0.05 is considered as statistical significance. *p<0.05, **p<0.01, ***p<0.001, ****p<0.0001. (**c**) Boxplot showing the relative mRNA expression of *Mapk8*, *Mapk9*, and *Mapk10* in the studied tissues. The count-based abundance of genes was transformed using the vst function and the Benjamini-Hochberg adjusted *p*-value (p.adj) is derived from DESeq2 (*Love et al., 2014*). *p.adj <0.05, **p.adj <0.01, ***p.adj <0.001, ****p.adj <0.0001. (**d**) Correlation between the relative mRNA expression of *Mapk8* and (**e**) *Mapk9* in skeletal muscle tissue (SkM) and plasma triglycerides level.

*and Frye, 2007*), an assay for assessing anxiety-like behavior in rodents, showed that rats in all groups had comparable open-arm entries and durations (*Supplementary file 5*). Collectively, the behavior tests indicate the JNK-IN-5A-treated rats exhibit no evidence of anxiety and behavior disorders.

To investigate the gene-level inhibition of JNK upon JNK-IN-5A treatment, we further examined the mRNA expression of JNK1 (encoded by the gene *Mapk8*), JNK2 (encoded by the gene *Mapk9*), and JNK3 (encoded by the gene *Mapk10*) in the study. As anticipated, *Mapk8* and *Mapk9* have high expression in all these tissues (liver, vWAT, SkM, and brain) whereas *Mapk10* is highly expressed specifically in the brain tissue (*Figure 2c*, *Figure 3—figure supplement 1a*). Our analysis indicates that the treatment of JNK_D1 effectively alleviated the JNK1 gene expression in the liver (padj = 0.03) and exerted more profound inhibitory effects on both JNK1 (padj = 0.0009) and JNK2 gene expression (padj = 2.2e-07) in SkM (*Figure 2c*, *Supplementary file 6*). Interestingly, we found that plasma TG level is significantly correlated with the JNK2 gene expression in SkM (*Figure 2d and e*). In brain tissue, we found JNK3 gene expression exhibited a trend of dose-dependently inhibition in the JNK-IN-5A-treated groups. Particularly, the downregulation of JNK3 in the sucrose + JNK_D2 group approaches the borderline of significance (padj = 0.07), compared to those, which drank sweetened water group only (sucrose group) (*Supplementary file 6*).

## JNK inhibition rewires metabolic perturbation in the liver and vWAT

To gain in-depth insight into system-level transcriptional response to sucrose consumption and JNK-IN-5A treatment, we next studied the global gene expression profiles in both liver and extra-hepatic tissues (vWAT, SkM, and brain). A total of 19,780 protein-coding genes in the rat genome were analyzed (*Martin et al., 2023*). Principal component analysis (PCA) revealed that liquid sucrose ingestion resulted in the largest transcriptional changes in the liver and vWAT (*Figure 3a*), which is consistent with previous results (*Stephenson et al., 2022*). Of note, we observed that JNK-IN-5A treatment, especially JNK_D1, corrected to a large extent of the gene expression profile in the liver and vWAT of treated-rats towards the profile of healthy controls on a principal components space (*Figure 3a*). In SkM, the transcriptional difference is primally driven by JNK_D1 treatment and showed a clear separation (*Figure 3a*).

In parallel with the results presented above, differential gene expression analysis revealed that the liver had the largest number of differentially expressed genes (DEGs, DESeq2-Benjamini-Hochberg adjusted p<0.01) associated with liquid sucrose ingestion, comprising a total of 2771 DEGs (14%), followed closely by vWAT with 2,724 DEGs (13.8%), SkM with 257 DEGs (1.3%), and the brain with a modest 30 DEGs (0.2%) (*Figure 3b and c*, *Figure 3—figure supplement 1b–e*, *Supplementary file 6*). Moreover, we observed that JNK_D1 treatment demonstrated a remarkable effect in the reversal of gene expression patterns related to sucrose consumption. Specifically, in the liver, JNK_D1 treatment substantially suppressed the expression of 60.7% (1143 out of 1884, hypergeometric p≈ 0) sucrose-induced DEGs and elevated the expression of 22% of sucrose-repressed genes (195 out of 887, hypergeometric p=6.9e-129) (*Figure 3d*, *Figure 3—figure supplement 1b*). Likewise, we observed a reversal of 42.2% and 31.4% (698 of 1655 and 336 out of 1069, respectively) DEGs in vWAT (*Figure 3d*, *Figure 3—figure supplement 1c*). It should be noted that, in both liver and vWAT, the effect of JNK_D2 treatment is relatively smaller than JNK_D1 in terms of the corrections of DEGs (*Figure 3—figure supplement 2a and b*). Pathway enrichment analysis of JNK_D1 reversed genes identified significantly enriched KEGG pathways related to carbon metabolism, fatty acid metabolism, protein processing in ER, and oxidative phosphorylation (OXPHOS) (Benjamini-Hochberg adjusted p<0.01) (*Figure 3—figure supplement 2c and d*). Interestingly, some of the genes exhibited opposing regulation in the liver and vWAT in response to the interventions of sucrose only and the combination of sucrose intervention and JNK-IN-5A treatment. This is exemplified in genes related to fatty acid

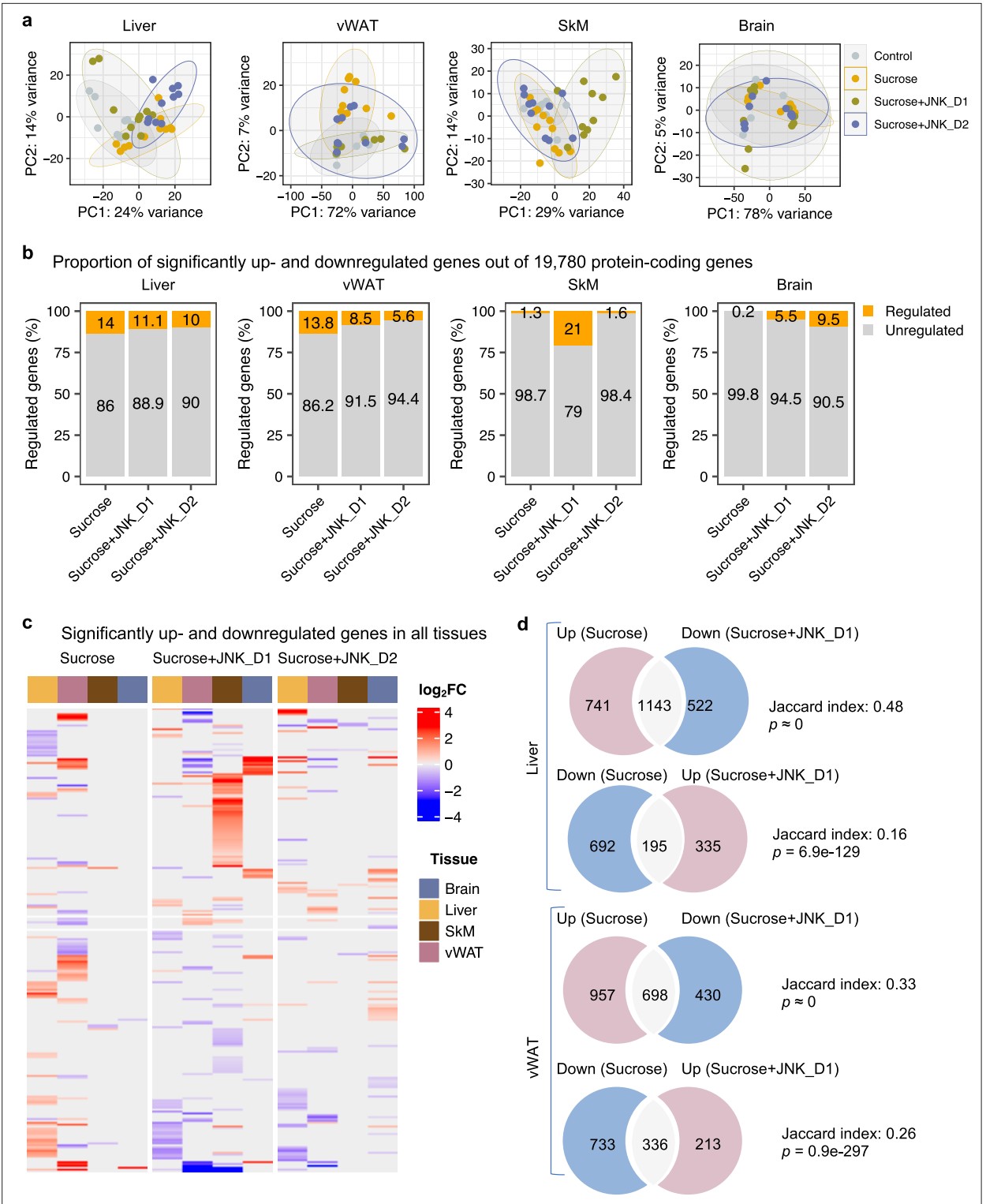

**Figure 3.** System-wide transcriptomics profiling revealed tissue-specific metabolic rewiring by sucrose consumption and JNK-IN-5A treatment. (**a**) Principal component analysis (PCA) of liver, visceral white adipose tissue (vWAT), skeletal muscle (SkM), and brain transcriptome data showing global gene expression profiles in control (n=9), sucrose (n=11), sucrose+JNK_D1 and sucrose+JNK_D2 (n=10/per group). Each data point represents a sample in the respective colored group. (**b**) Bar graphs represent the percentage of differential expressed genes (or regulated genes) associated with sucrose (sucrose *vs.* control), sucrose+JNK_D1 (sucrose+JNK_D1 *vs.* sucrose), or sucrose+JNK_D2 (sucrose+JNK_D2 *vs.* sucrose) in all protein-coding genes (N=19,780) for each tissue. (**c**) Heatmap of all genes differentially upregulated (red) and downregulated (blue) in three pairwise comparisons for sucrose (sucrose *vs.* control), sucrose+JNK_D1 (sucrose+JNK_D1 *vs.* sucrose), and sucrose+JNK_D2 (sucrose+JNK_D2 *vs.* sucrose) (adjusted p-value, p.adj<0.01)

*Figure 3 continued on next page*

*Figure 3 continued*

for each tissue. Column annotation represents tissues. (**d**) Venn diagram showing the overlap between significantly regulated genes response to sucrose consumption and JNK_D1 treatment.

The online version of this article includes the following figure supplement(s) for figure 3:

**Figure supplement 1.** Intersection between differentially expressed genes in different tissues.

**Figure supplement 2.** Transcriptional changes in the liver and adipose tissues in response to JNK-IN-5A treatment.

**Figure supplement 3.** Relative changes of genes involved in fatty acid degradation, elongation, biosynthesis, and metabolism in response to sucrose consumption and JNK-IN-5A treatment in the tissues.

**Figure supplement 4.** Relative changes of genes involved in fatty acid uptake in response to sucrose consumption and JNK-in-5A treatment in the tissues.

**Figure supplement 5.** Relative changes of genes involved in redox homeostasis in response to sucrose consumption and JNK-in-5A treatment in the tissues.

degradation, elongation, and biosynthesis (up in the liver and down in vWAT in sucrose only group, while down in the liver and up in vWAT in sucrose plus JNK-IN-5A-treated groups), including *Cpt2, Acads, Acadvl, Acat1, Acot5, Eci3, Hsd17b12,* and *Oxsm* (*Figure 3—figure supplement 3*, *Supplementary file 6*); genes related to fatty acid (FA) uptake, including fatty acid binding proteins *Fabp1, Fabp4,* and transporters *Slc27a1* and *Slc27a5* (*Figure 3—figure supplement 4*, *Supplementary file 6*); and genes related to redox homeostasis, including *Prdx4,* a secreted enzyme that scavenges redox oxidative stress (ROS), *Selenos,* and *Ndufa6* (*Figure 3—figure supplement 5*, *Supplementary file 6*). Taken together, these results suggest that JNK-IN-5A treatment reshapes sucrose-induced transcriptional changes in both the liver and vWAT, reinforcing their metabolic synergy in FA uptake and metabolism (*Stephenson et al., 2022*).

## JNK inhibition in SkM correlated with the regulation of energy metabolism in the liver and vWAT

It has been reported that JNK acts as a critical mediator of muscle phenotype and adaptive remodeling in animals and humans (*Lessard et al., 2018*). Here, we found that in contrast to the massive transcriptional regulation observed in the liver and vWAT, liquid sucrose intake has minimal impact on SkM in terms of both genome-wide transcriptional variance (*Figure 3a*) and the number of DEGs (*Figures 3a and 4a and b*). While in JNK_D1-treated SkM, a substantial quantitative difference in gene expression emerged, encompassing 4,159 (20.9%) DEGs, of which 97% up-regulated (1552 out of 1588) and 99% down-regulated (2557 out of 2571) genes being exclusively dependent on the JNK_D1 treatment (*Figure 4a and b*). Functional analysis revealed that many of the upregulated genes are involved in immune-related pathways, including hematopoietic cell lineage (HSCs), osteoclast differentiation, chemokine signaling pathway, phagosome, and Fc gamma R-mediated phagocytosis, and down-regulated genes are involved in OXPHOS, thermogenesis, TCA cycle, carbon metabolism, cardiac muscle contraction, and insulin signaling pathway (*Figure 4c and d*, *Figure 5*). Among the genes regulated by both sucrose-feeding and JNK_D1 treatment in SkM were those involved in the insulin signaling pathway (e.g. *Ppp1r3f, Phkg1, Phkb*) (*Figure 5*). As this intra-tissue effect was observed in JNK_D1 not in the JNK_D2-treated group (*Figure 3b*, *Figure 4—figure supplement 1*), we subsequently investigated the inter-tissue regulation based on the significant transcriptional changes (as indicated by the $\log_2$-Fold change, $\log_2$FC) in response to JNK_D1 treatment and found that $\log_2$FC of shared DEGs between JNK_D1-treated SkM and liver, vWAT, or brain exhibited strong agreements (*Figure 4e*, *Figure 4—figure supplement 1*). Specifically, 86% and 69% of shared DEGs between SkM and vWAT, and the brain, exhibited consistent changes in response to JNK_D1 treatment (the same $\log_2$FC sign, indicating either upregulated or downregulated in both tissues being compared), with a Spearman correlation of 0.78 and 0.54, respectively. These shared DEGs were found to be involved in several critical metabolic processes, including cholesterol metabolism, bile acid biosynthesis and secretion (such as *Lipc, Apob, Slc7a1, Angptl3, Slc10a1*), and regulation of energy homeostasis (e.g. *Lep*), suggesting that SkM had a major metabolic adaptation to JNK_D1 treatment and exerts its metabolic effect, at least partially, through coordinated interactions with other tissues.

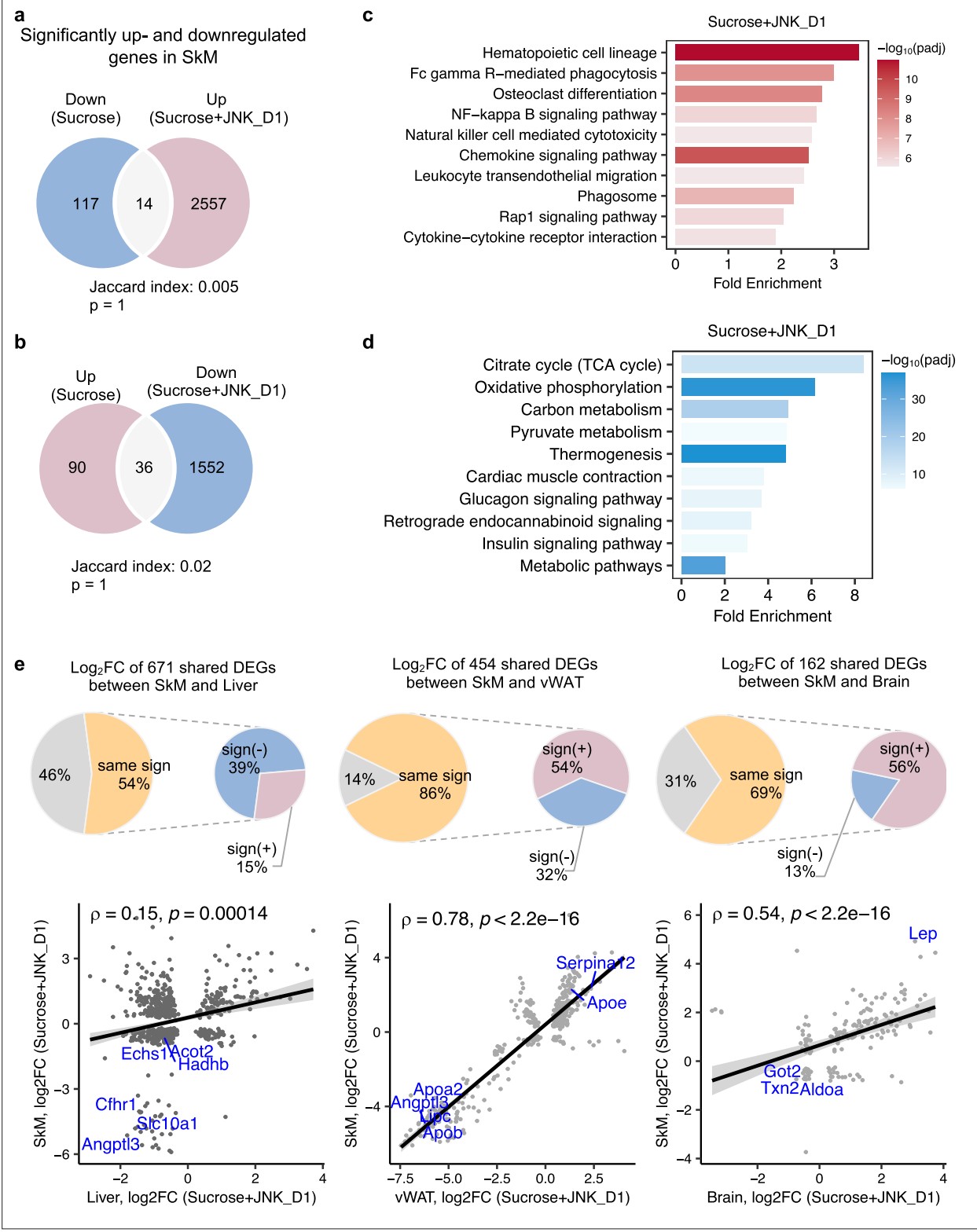

**Figure 4.** c-Jun N-terminal kinase (JNK) inhibition in skeletal muscle (SkM) correlated with the regulation of energy metabolism in the liver and adipose tissues. (**a, b**) Venn diagram showing the overlap between significantly regulated genes response to sucrose consumption and INK_D1 treatment in SkM. (**c, d**) Functional annotation of up- (red) and down-regulated (blue) genes in JNK_D1-treated SkM with a Benjamini Hochberg adjusted p-value <0.05. (**e**) Correlation of log₂(Fold change) for shared differentially expressed genes (DEGs) between the SkM and Liver, visceral white adipose tissue (vWAT), or Brain after JNK_D1 treatment.

*Figure 4 continued on next page*

*Figure 4 continued*

The online version of this article includes the following figure supplement(s) for figure 4:

**Figure supplement 1.** Transcriptional changes in response to JNK-IN-5A treatment.

## Inter-tissue network analysis revealed the JNK-IN-5A inhibition-associated molecular mechanism

To better understand the regulation of lipid metabolism and energy homeostasis at a system-wide level in an unbiased manner, we next performed gene co-expression network (GCN) analysis to study intra- and inter-tissue coordination of metabolic pathways as well as biological processes that are associated with sucrose-feeding and/or JNK_D1 inhibition. GCN, a robust systems biology approach (*Barabasi and Albert, 1999*), allows us to study the functionality of the gene module and its constituent genes. These genes are more likely to be co-expressed if they are either regulated by the same transcription factors or share similar topological properties in protein-protein interaction networks (*Califano et al., 2012*; *Dobrin et al., 2009*; *Zhang et al., 2016*). By investigating the importance and topology of these functional modules, it is possible to understand the biological mechanisms underpinning the disease and/or therapy intervention (*Choobdar et al., 2019*; *Yang et al., 2021*; *Yang et al., 2014*).

We first constructed tissue-specific GCN based on the whole transcriptomics data and selected highly connected genes from the network (the top 10% positively correlated genes that fulfilled FDR <0.05), followed by module detection using the Leiden clustering algorithm (*Figure 6—figure supplement 1*, *Supplementary file 7*). We subsequently evaluated the enrichment of DEGs in the modules to identify those associated with sucrose consumption, JNK_D1 treatment, or both (so-called perturbated module). As expected, GCN analysis revealed distinct gene expression patterns across different tissues (*Figure 6a*). For instance, in module 1 in the liver (Liver.M1), vWAT.M4, SkM.M4, and Brain.M4, we found a significant enrichment of genes that exhibited elevated expression in response to sucrose consumption while simultaneously being suppressed by JNK_D1. Conversely, Liver.M0 and vWAT.M3 exhibited significant enrichment of genes repressed by sucrose intake but elevated after JNK_D1 treatment. These distinct expression patterns collectively reflect the combined metabolic effect of sucrose-feeding and JNK_D1 interventions. Additionally, GCN analysis identified modules associated with either sucrose-feeding or JNK_D1 intervention at the tissue level, for example, vWAT.M0 (sucrose), vWAT.M1 and Brain.M5 (JNK_D1) (*Figure 6a*).

To pinpoint gene modules involved in inter-tissue communication, we conducted a correlation analysis among inter-tissue modules using their module eigengene (see details in Method) and assessed the enrichment of genes from these modules in MSigDB hallmark gene sets. Our analysis revealed a significant correlation between SkM.M0 and Liver.M0 ($R = 0.67$, padj = 0.006) as well as vWAT.M3 ($R = 0.58$, padj = 0.03) (*Figure 6b and c*). This correlation aligns with the functional annotation of genes within these modules, including those associated with mTORC1 signaling, PI3K/AKT/mTOR signaling, DNA Repair, and unfolded protein response (*Figure 6d*). Interestingly, we observed that *Mapk9* is among the module members of SkM.M0, indicating a potential metabolic link coordinated by JNK inhibition, connecting SkM with other tissues, including the liver and vWAT (*Figure 6b*, *Supplementary file 7*). SkM.M1 and Liver.M2 ($R = -0.66$, padj = 0.006), SkM.M3 and vWAT.M0 ($R = 0.60$, padj = 0.03), and SkM.M4 and vWAT.M3 ($R = -0.66$, padj = 0.006) were also found to be significantly correlated ($R = -0.57$, padj = 0.03). Note that genes in Liver.M1, vWAT.M3, SkM.M0, SkM.M4, and Brain.M4 are commonly involved in adipogenesis, fatty acid metabolism, OXPHOS, and reactive oxygen species pathway (*Figure 6d*). These observations suggest the potential crosstalk or regulatory relationship between these tissue modules after JNK_D1 treatment.

## Metabolic modeling identified the metabolic reprogramming linked to JNK-IN-5A inhibition

To predict how tissue-level metabolic activity responds to sucrose feeding and JNK-IN-5A treatment, we performed an integrative analysis of transcriptomics data by generating tissue-specific GEMs and applying Compass (*Wagner et al., 2021*). Compass is a flux balance analysis-based (*Orth et al., 2010*) algorithms to model the metabolic state of a system (e.g. single cell or tissue), taking into account the observed expression levels of enzyme-coding genes in each sample from RNA-seq data. In this study,

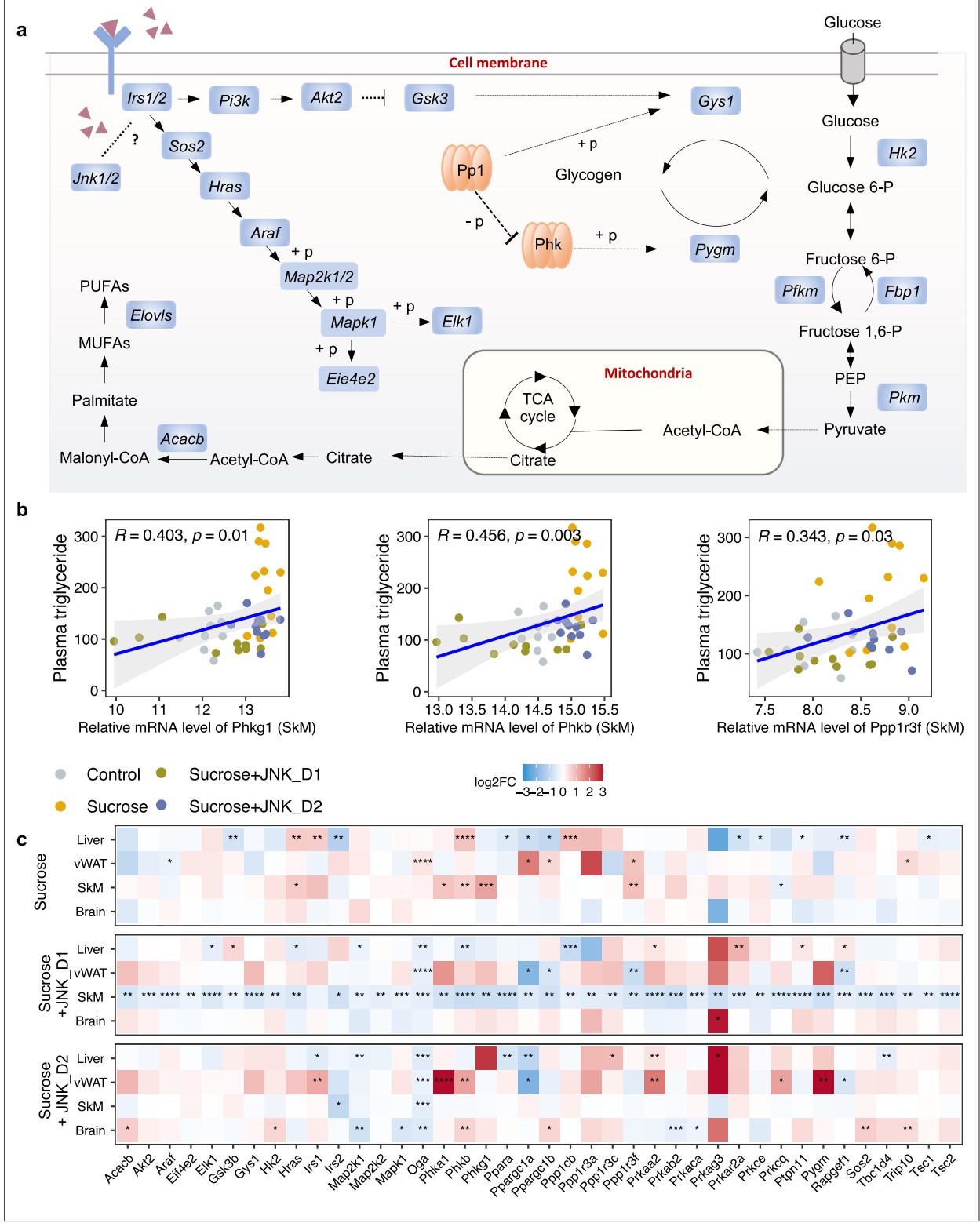

**Figure 5.** c-Jun N-terminal kinase (JNK) inhibition regulates insulin signaling-related genes in skeletal muscle (SkM). (**a**) A representative diagram of metabolic pathways associated with insulin resistance and insulin signaling was found to be largely inhibited by JNK_D1 treatment. (**b**) Correlation between the relative mRNA expression of *Phkg1, Phkb, and Ppp1r3f* (those three genes are also significantly upregulated in sucrose-feeding only group) in SkM and plasma triglycerides level. (**c**) Relative changes of genes involved in insulin resistance and insulin signaling pathways in response to sucrose consumption and JNK-IN-5A treatment in the tissues. The BH-adjusted p-value (p.adj) is derived from DESeq2 (*Love et al., 2014*). *p.adj<0.05, **p.adj<0.01, ***p.adj<0.001, ****p.adj<0.0001.

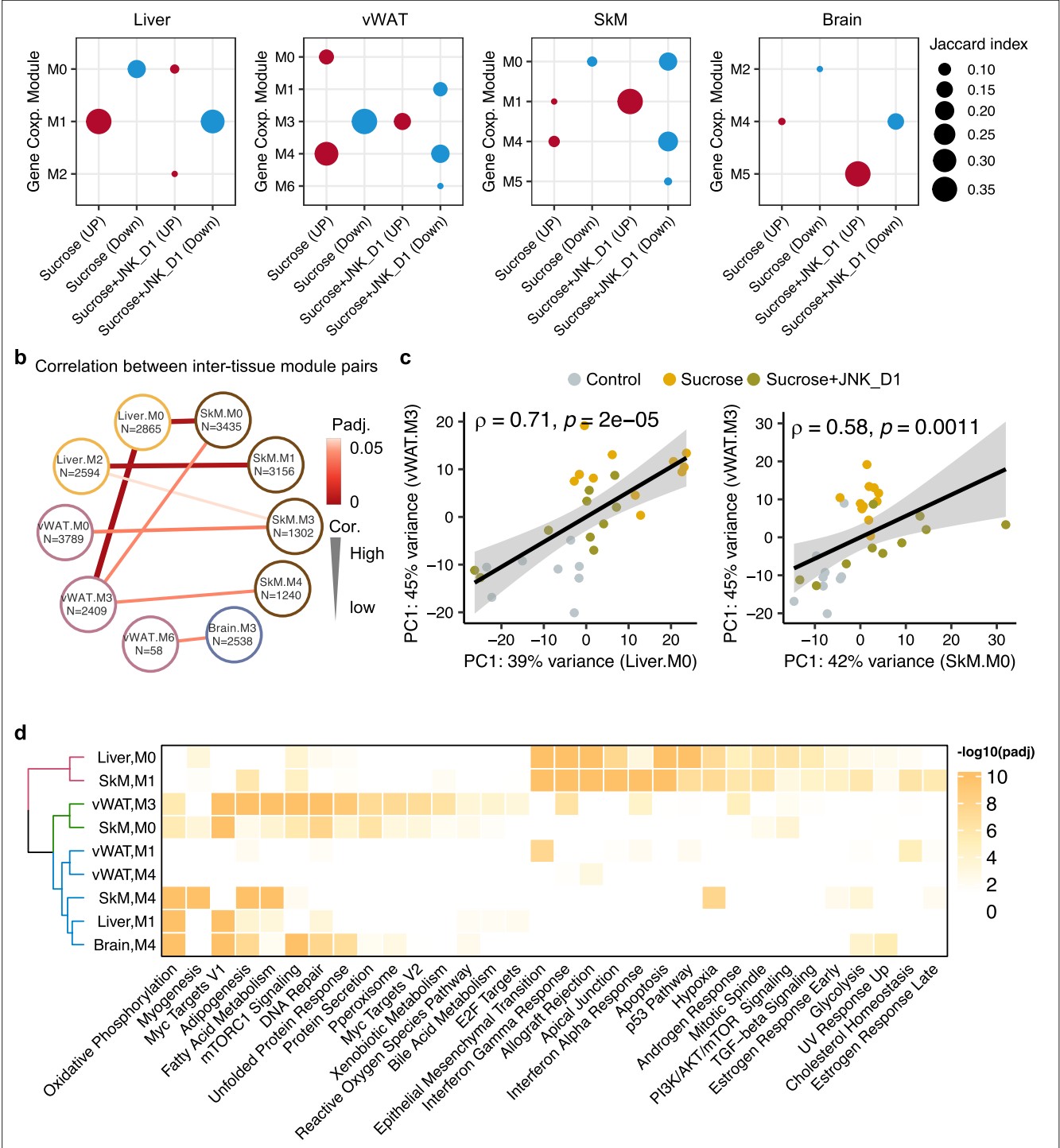

**Figure 6.** Inter-tissue network analysis identifies JNK-IN-5A inhibition-associated molecular mechanism. (**a**) Significantly enriched modules (hypergeometric test, <0.05) by the differentially expressed genes related to sucrose, sucrose+JNK_D1, and sucrose+JNK_D1 in each tissue. (**b**) Dot-plot showing the significant correlation among inter-tissue module pairs. The size and color of the connected line are proportional to the correlation coefficient and statistical significance indicated by the adjusted p-value. The module pairs with Benjamini Hochberg adjusted p-value (p.adj)<0.05 are presented. (**c**) Spearman coefficient correlation of the first component from principal component analysis (PCA) analyses on gene expression of module pairs. (**d**) Significantly enriched MSigDB hallmark gene sets by each module.

The online version of this article includes the following figure supplement(s) for figure 6:

**Figure supplement 1.** Tissue-specific gene co-expression network analyses.

we employed Recon3D, a generic GEMs of human metabolism, comprising 10,600 metabolic reactions associated with 5835 metabolites and 2248 metabolic genes (*Brunk et al., 2018*). Consistent with our previous analysis, we observed a shift in metabolic activity in the liver and vWAT after JNK_D1 treatment (*Figure 7*, *Supplementary file 8*). Specifically, Compass predicted that 2840 and 4170 metabolic reactions showed significantly altered activity (Wilcoxon test, adjusted p-value <0.05) following the sucrose consumption in the liver and vWAT, respectively. 89.2% and 88.3% of which were correspondingly reversed after JNK_D1 treatment (*Supplementary file 9*). The common altered reactions are partitioned into the subsystems named citric acid cycle, fatty acid oxidation, fatty acid synthesis, and tryptophan metabolism in Recon3D (*Figure 7c*). Some of the predicted metabolic changes align with previous findings in rodents subjected to sucrose overconsumption. For example, Öztürk et al. reported altered tryptophan metabolism in rats consuming 10% sucrose in drinking water, including decreased serum levels of kynurenic acid and kynurenine (*Ozturk and Eryavuz Onmaz, 2022*). Similarly, increased triglyceride-bound oleate, palmitate, and stearate were observed in the livers of rats fed a 10% sucrose solution (*Stephenson et al., 2022*), indicating JNK-IN-5A treatment may regulate lipid metabolism by modulating these metabolic activities. Moreover, we found 8 metabolic reactions significantly altered in sucrose +JNK_D2-treated rats, including 2 reactions involved in glycolysis/gluconeogenesis, 2 in nucleotide interconversion, 1 in pentose phosphate pathway, 1 in fructose and mannose metabolism, 1 in propanoate metabolism (Cohen's d<0, padj <0.05), and 1 in drug metabolism (Cohen's d>0, padj <0.05) (*Supplementary file 8*). Hence our systems biology analysis revealed the molecular mechanisms altered due the JNK-IN-5A treatment.

## Discussion

The etiology of MAFLD is multifactorial and remains incompletely understood, impeding the progress in developing effective therapies for MAFLD. Our investigation into the metabolic changes from the major metabolically active tissues enabled us to systematically assess the potential therapeutic effects of JNK-IN-5A under high sucrose consumption. Here, we demonstrated that JNK-IN-5A attenuated the accumulation of circulating TG and inflammation exaggerated by liquid sucrose consumption. Importantly, the rats treated with JNK-IN-5A did not exhibit indications of anxiety or behavioral disorders. These results have important implications for the development of JNK3 inhibitors, taking into account the high expression of JNK3 in the brain (*Chang and Karin, 2001*). Additionally, our data indicated that JNK-IN-5A, a selective inhibitor of JNK2 and JNK3 (*Angell et al., 2007*; *Cerbone et al., 2012*), also exerted an inhibitory effect on the gene expression of JNK1 in in vivo, especially in skeletal muscle (*Figure 2*).

By studying the hepatic and extrahepatic tissues using genome-wide transcriptome data and integrative analyses with GEMs, we found that the intervention with sucrose-sweetened water triggered extensive transcriptional changes in the liver and adipose tissues, including those involved in fatty acid and oxidative metabolism. The metabolic axis between the liver and adipose tissues is a critical regulatory network that governs energy balance and metabolic homeostasis in the body (*Azzu et al., 2020*). It has been reported that upon chronic overnutrition, adipose lipogenesis is significantly downregulated, which leads to ectopic lipid accumulation and insulin resistance (*Jeon et al., 2023*). In line with this, we found that in the sucrose-sweetened water group, the mRNA expression of many genes related to fatty acid metabolism and uptake is downregulated in adipose tissue. Interestingly, those genes showed the opposite regulation between liver and adipose tissue, indicating their complementary role in these metabolic processes in response to the diet intervention. These findings also pinpoint the critical role of extrahepatic metabolism in the development of MAFLD.

JNK plays essential roles in a range of signaling pathways that have implications for metabolic processes such as insulin sensitivity, thermogenesis, adipogenesis, and lipid metabolism (*Nikolic et al., 2020*; *Solinas and Becattini, 2017*). Our data demonstrated that JNK-IN-5A, especially when administered at a regular dosage (JNK_D1, 30 mg/kg/d), effectively opposite a substantial portion of the transcriptional changes induced by sucrose consumption, particularly those related to carbon, fatty acid metabolism, OXPHOS, and thermogenesis in the liver and adipose tissues. This suggests that JNK-IN-5A might hold promise as a potential strategy for restoring metabolic perturbation in tissues challenged by sucrose-induced stress. However, our study also revealed that JNK_D1-treated skeletal muscle exhibited profound transcriptional changes related to HSCs, osteoclast differentiation, and immune-related pathways. Importantly, we found that these changes were not confined to

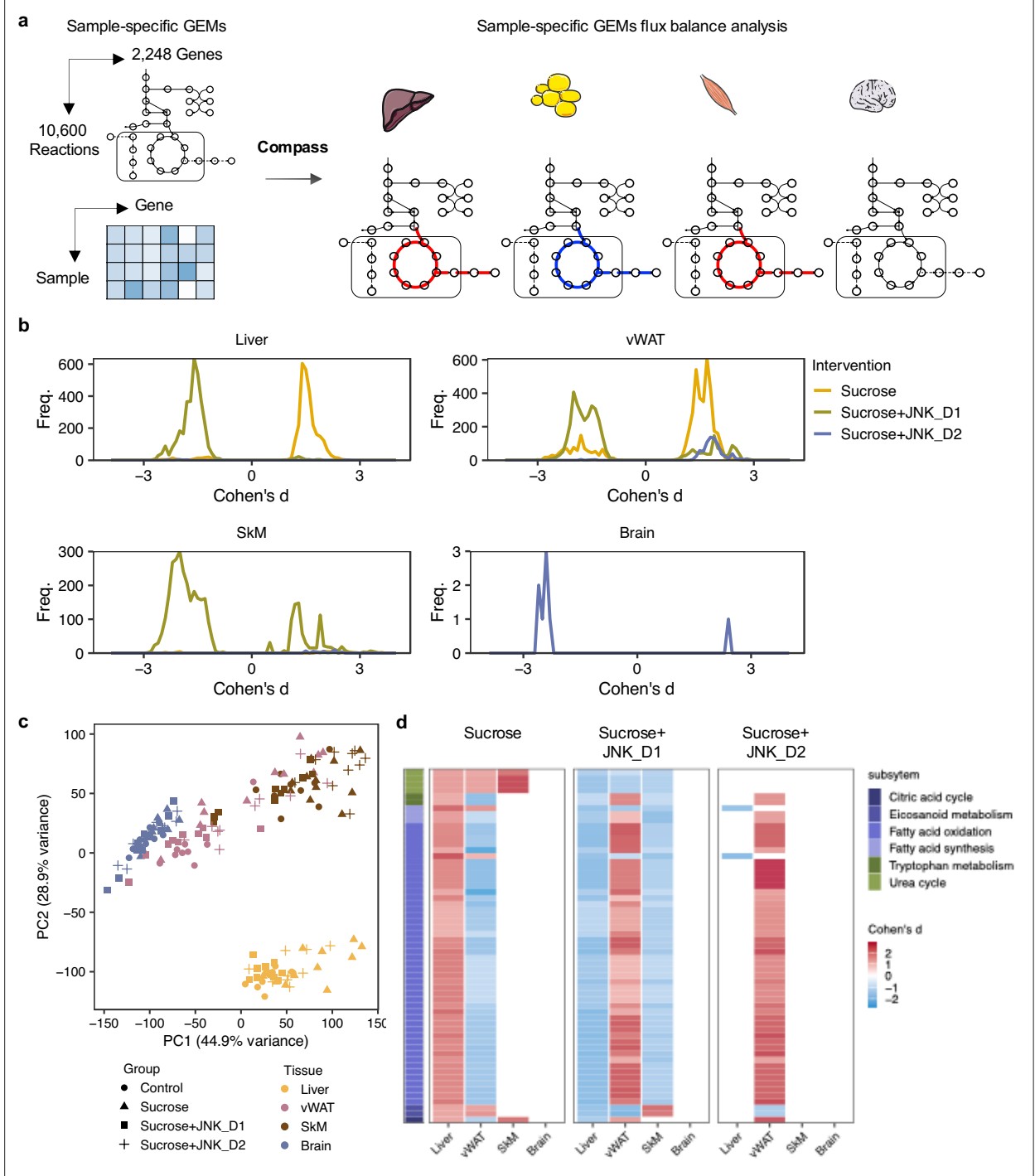

**Figure 7.** Metabolic modeling highlights metabolic reprogramming linked to JNK-IN-5A inhibition. (**a**) Simulation of metabolic activity based on Compass (**Wagner et al., 2021**) and Recon3D (**Brunk et al., 2018**). (**b**) Distribution of Cohen's d statistic for differential metabolic reactions in response to sucrose, sucrose+JNK_D1 and sucrose+JNK_D2 in each tissue, see also **Supplementary file 8**. (**c**) Principal component analysis of the Compass score of metabolic reaction potential activity scores showing the metabolic heterogeneity in response to sucrose ingestion and JNK-IN-5A treatment at the tissue levels, for principal components 1 and 2. (**d**) Heatmap showing the tissue-level differential activity of metabolic reactions in response to sucrose intake and JNK-IN-5A treatment. Reactions are partitioned by Recon3 pathways (**Brunk et al., 2018**) and colored by the sign of their Cohen's d statistic derived from different contrasts: sucrose *vs* control, sucrose+JNK_D1 *vs* sucrose, or sucrose+JNK_D2 *vs* sucrose, respectively. Abbreviations: vWAT, visceral white adipose tissue; SkM, skeletal muscle.

skeletal muscle but exhibited correlations with transcriptional changes in the liver, adipose tissue, and even brain tissue. The suggest the coordinated roles of JNK inhibition in mitigating sucrose-induced MAFLD across these tissues, while the exact molecular mechanisms by which JNK-IN-5A exert its muscle-specific effect warrant further investigations. Previously, *Xiao et al., 2019* demonstrated that the JNK pathway regulates HSCs expansion and self-renewal in JNK-IN-8-treated CD34+CD45RA-CB cells. *Lessard et al., 2018* demonstrated that mice with muscle-specific JNK1/2 knockout, have a muscle phenotype that mimics that of endurance-trained athletes. Together with our data, these results suggest a multifaceted role of JNK signaling in the regulation of HSCs and metabolism in skeletal muscle.

Although overactivated JNK activity presents an attractive opportunity to combat MAFLD, inhibition of JNK presents substantial challenges and potential risks due to its broad and multifaceted roles in many cellular processes. One key challenge is the dual role of JNK signaling (*Lamb et al., 2003*). For instance, long-term JNK inhibition may disrupt liver regeneration, as JNK plays a critical role in liver repair by regulating hepatocyte proliferation and survival following injury or stress (*Papa and Bubici, 2018*). In HCC, it has been reported that JNK acts as both a tumor promoter, driving inflammation, fibrosis, and metabolic dysregulation, and a tumor suppressor, facilitating apoptosis and cell cycle arrest in damaged hepatocytes. Its inhibition, therefore, carries the risk of inadvertently promoting tumor progression under certain conditions (*Seki et al., 2012*). Furthermore, the differential roles of JNK isoforms (JNK1, JNK2, JNK3) and a lack of specificity of JNK inhibitors present another layer of complexity. Given these challenges, while our study demonstrated the potential of JNK-IN-5A in mitigating early metabolic dysfunction in the liver and adipose tissues, JNK targeting strategies should be carefully tailored to the disease stage under investigation. For curative approaches targeting advanced MAFLD, such as MASH, future studies are warranted to address considerations related to dosing, tissue specificity, and the long-term effects.

In summary, our study provides a comprehensive understanding of the therapeutic potential of JNK-IN-5A treatment in a rat model of sucrose-induced MAFLD. Through an analysis of genome-wide transcriptome data in the major metabolic tissues, we shed light on the JNK-IN-5A-mediated metabolic effects in both hepatic and extrahepatic tissues. This implies a potential therapeutic strategy for the treatment of diet-induced MAFLD and other metabolic complications, potentially by regulating cellular energy homeostasis and mitochondrial metabolism. Nevertheless, several limitations warrant consideration. First, while we observed transcriptional adaptations in skeletal muscle tissue following treatment, the exact molecular mechanisms underlying these changes and their roles in skeletal muscle function and systemic metabolic homeostasis remain unclear. Further investigation is warranted to elucidate the muscle-specific effects of JNK inhibition. Second, our study did not investigate the dose-dependent or potential off-target effects of JNK-IN-5A, particularly its activity on other members of the kinase family and associated signaling pathways. Lastly, the long-term effects of JNK-IN-5A administration remain unexplored. Understanding its prolonged impact across different stages of MAFLD, including advanced MASH, is crucial for assessing the full therapeutic potential of JNK inhibition in the treatment of MAFLD.

## Materials and methods
### Experimental models and subject details
#### Oral toxicity study in rats

Twelve male and twelve female rats of the Wistar strain (Envigo, Venray, Netherlands) were used to assess the 7 d oral (gavage) toxicology, in terms of clinical signs, body weight, and clinical pathological parameters, of JNK-IN-5A in rats. Animals were group-housed under standard conditions and had free access to water and 2916 C Teklad global 16% protein rodent irradiated diet (Envigo, Venray, Netherlands). Dose levels of 0 (vehicle control), 30, 100, or 300 mg/kg of JNK-IN-5A were administered once daily via oral gavage for 7 d. The vehicle was 2 w/w% HPMC (Colorcon), 2 w/w% PS80 (Merck) in 10 mM PBS, pH 7 and dose volume was 5 mL/kg for JNK-IN-5A. Body weights were recorded the week before and within 24 hr before the start of dosing and on Day 3 and on necropsy Day 8. All animals were bled at 2 hr after dosing on Day 1 and Day 7. Blood samples were placed on ice after sample collection and centrifuged at approx. 3000×G, 5 min., 4 °C. 7.5 µL plasma was transferred into a separate low-binding protein tube and the remaining plasma was transferred into a second

low-binding tube. The tubes were frozen upright at approximately –20 °C. Blood and plasma were analyzed using an Exigo analyzer and Vetscan equipment. The VetScan analyzer was calibrated with calibration samples with known plasma levels of each parameter before the start of analysis, occasionally during the analysis period of study samples, and after completed analyses. All the animal procedures (including housing, health monitoring, dosing, etc.) and ethical revisions were performed according to the 2010/63/EU Directive on the protection of animals used for biomedical research.

## JNK-IN-5A treatment study in rats

All experiments were performed in accordance with institutional ethical guidelines, and the present in vivo study was approved by the Ethics Committee for Animal Research of Atatürk University (with the reference number 2023/04 on 30.03.2023). Rats were obtained from the Atatürk University Experimental Research Center. All groups were housed on a 12 hr light and dark cycle. The temperature and humidity values of the environment were kept at optimal values. To prevent neophobia, rats were divided into groups 1 wk before sucrose treatment by following the double-blind method to adapt to the laboratory environment and each other.

A total of 4 groups consisting of 11 subjects each were formed, including the healthy Control group consisting of rats received ad libitum tap water, the Sucrose group consisting of rats received ad libitum 10% Sucrose (BioShop, Canada, Lot No: 2E77269) (equivalent to the concentration of fructose in typical soda *Jang et al., 2020*), and groups treated by JNK-IN-5A at two concentration: sucrose+JNK_D1 group with rats received ad libitum 10% sucrose and 30 mg/kg/d JNK-IN-5A gavage and sucrose+JNK_D2 group with rats received ad libitum 10% sucrose and 60 mg/kg/d JNK-IN-5A gavage. The rats had ad libitum access to tap water (Control) or 10% sucrose water (sucrose, sucrose+JNK_D1, and sucrose+JNK_D2) and a standard chow diet throughout the experimental period. The amount of water consumed by all groups was recorded daily, and drinking water was renewed every day. After feeding with sucrose for 4 wk, drug treatment was given at the following 3 wk. Behavioral tests were taken before the experiment started and after 3 wk of drug treatment. Locomotor activity tests, elevated plus maze tests, and passive avoidance tests were used as behavioral tests. Behavioral tests were repeated at the end of the 21 d treatment regimen. While the last behavioral tests were being taken, drug treatment continued so that the drug level in the blood would not change. After the experimental animals were sacrificed with a high dose (200 mg/kg thiopental sodium) anesthetic, the relevant tissues were taken for further analysis.

## Monitoring behavior

All tests were performed twice in total, initially and after the 21 d treatment regimen. According to the results obtained from the initial measurements, rats with outliers for the experiment were determined. After the treatment measurement, it was determined whether there was any change in the behavior of the rats. The Locomotor Activity Test is a system used to test differences in motor activity and anxiety-related behaviors of rats. In the system scanned with infrared rays, each of the ambulatory, stereotypical, rearings, resting percentage and traveled distance movements of the rats are digitized and converted to computer output. These outputs are evaluated and provide information that needs to be interpreted about the animal's adaptation to the new environment and the continuation of the sense of curiosity or anxiety. Stereotypical movement shows rats grooming and recognizing their environment. Ambulatory movement is all the movements that rats make on the ground without standing up. Distance shows the distance traveled by the rats in the cage. The rearing movement indicates the time the rats stand up in the cage. The resting percentage is an indicator of anxiety. Ambulatory movements are recorded with horizontal sensors that surround the cage 360 degrees at the base, and stereotypic and rearing movements are recorded with vertical sensors located 10 cm above the floor. The entire cage continues to be measured continuously with 250 ms infrared. Unlike the open field test, activity measurement is performed in this study and the computer automatically records all the recordings instead of an independent observer (*Lu et al., 2019*). The experiment is performed in a dark environment. Rats are placed in cages for 10 min periods and the system automatically records with infrared rays. After each rat, the cage is cleaned with alcohol to remove odor and urine (*Ferah Okkay et al., 2022*). The Elevated Plus Maze Test is used to measure anxiety-like behavior in rodents. The elevated plus maze apparatus is a system consisting of two open arms (without walls; 50 cm × 10 cm) and two closed arms (with black walls, 50 cm × 10 cm). The open and closed arms are connected

by a central platform. The labyrinth is 55 cm above the ground. At the start of each test, rats are placed in the central zone and allowed to explore the platform for 5 min. The rats are recorded with a camera located at the top of the maze and the test is done in a quiet, dim room. The number of times the rats entered the open and closed arms, and the time spent in these arms were recorded. The device was cleaned with an alcohol solution between tests (*Chellian et al., 2020*; *Walf and Frye, 2007*). Results are given as arm entries and time spent (durations) in these areas. The more entries into open arms and the more duration spent there, the less anxiety the rat has. The Passive Avoidance Test is a system consisting of two rooms (light/dark), the base of which is covered with rods that can apply electroshock, in which memory, behavior, and learning abilities are investigated. This system is used to test the learning ability of the animal by fear-driven conditional avoidance. In this test, the animal is expected to learn to avoid entering the electroshock chamber (dark room). Rats are nocturnal animals. By nature, they sleep during the day and are active at night. Although rats are not afraid of light, they generally live in dark compartments because they are uncomfortable with the bright environment. In the passive avoidance system, lighting the bright compartment with LED lights and high lumen light creates an uncomfortable environment for the animals, and they want to move to the closed compartment. Here, the low intensity electroshock they are exposed to causes a deterrent stimulus and allows the rats to understand and learn how the system works. The experimental procedure is built on these understanding and learning abilities (*Giménez De Béjar et al., 2017*; *Hosseini et al., 2013*; *Wu et al., 2020*). The experiment is carried out on two consecutive days. Learning is measured on the first day and recall on the second day. The time that each rat took to enter the dark compartment was considered as initial latency. The next day (retention phase), the rats were reintroduced into the bright chamber and the step-through latency was noted down as memory retention. Shorter latencies indicated poorer cognition. At this stage, rats are expected to remember the shock in the dark compartment. For this reason, it was accepted that the memory of the rats that stayed longer in the bright compartment was stronger. In retention measurements, it is understood that the longer the rats stay in the bright compartment, the stronger the memory and learning.

## RNA extraction and library preparation for transcriptome sequencing

Total RNA is extracted from the snap-frozen rat tissues with Zymo Research Quick RNA Miniprep kit (California USA) by following the protocol provided on the manufacturer's website with slight modifications made to improve the quality and quantity of RNA from each tissue. Before the cell lysis, the tissues are subjected to short (1 min or longer for muscle tissues) homogenization with the beads in PowerLyzer 24 by keeping samples on ice to prevent overheating. The homogenized tissues are further enzymatically in a digestion buffer for at least 30 min or more for muscle tissue at room temperature with continuous inversion and agitation of the tubes. After the lysis of the cells, the RNA is cleaned and collected with columns. The quality and quantity of the isolated samples were examined spectrophotometrically with NanoDrop (Thermo Fisher, USA). After the evaluation of the sample A260, A260/A280, and A260/A230 values, the RNA integrity number (RIN) value of total RNA samples was measured with TapeStation (Agilent Tech, USA). The amount of total RNA was measured more precisely fluorometrically with the RNA Broad Range kit (Thermofisher, USA) by using Qubit (Invitrogen, USA). RNA sequencing library was prepared with Illumina Stranded Total RNA Prep and Ligation with Ribo-Zero Plus kit was used by following the standard protocol provided by the manufacturer. The libraries were then pair-end (2×100 bp) sequenced on the NovaSeq6000 system yielding, on average, 25 million fragment reads per sample. Raw sequencing data (.bcl) was converted to FASTQ with the Dragen Bio-IT platform (v3.9.5). The quality of RNA-seq data was assessed by FastQC (v0.11.9).

## RNA-seq data processing, differential expression, and functional enrichment analysis

Tissue bulk RNA-seq data were aligned and quantified using a standard protocol of Kallisto (v0.46.2) (*Bray et al., 2016*) against the *Rattus norvegicus* genome (mRatBN7.2) downloaded from Ensembl (*Martin et al., 2023*) official website (https://www.ensembl.org/index.html). The output of Kallisto, both estimated counts and TPM (transcript per kilobase million)-based transcript-level expressions were then transformed into gene-level expressions using the Bioconductor package tximport (v1.22.0) with the tx2gene option set to connect transcripts to genes (*Soneson et al., 2015*). Protein-coding

genes were considered for the above step and downstream analyses. Differential analysis was performed using the DESeq2 Bioconductor package (v1.34.0) (*Love et al., 2014*), following a standard protocol for all three pairwise intervention comparisons for determining the effect of sucrose consumption only (sucrose *vs* control), the effect of 30 mg/kg/d JNK-IN-5A treatment (sucrose+JNK_D1 *vs* sucrose), and effect of 60 mg/kg/d JNK-IN-5A treatment (sucrose+JNK_D2 *vs* sucrose) in all four studied tissues. Significantly expressed genes (DEGs) were identified with a significance threshold of an adjusted <i>p-value <0.01. To examine differences in global gene expression, we conducted principal component analysis (PCA) based on variance-stabilizing transformation (VST)-normalized count data in DESeq2. PC1 and PC2 were plotted for visualization.

### Gene co-expression network analysis

For the generation of the tissue-specific co-expression networks, we first filtered out lowly-expressed genes based on their TPM-based expression level (TPM <1) and constructed co-expression networks by generating gene pairs Spearman correlation ranks within the tissue, which was performed using the 'spearmanr' function from *SciPy* (*Virtanen et al., 2020*) in Python 3.8. Next, considering the network with negative correlation has relatively low correlation scores, we retained the top 10% positively correlated genes that fulfilled FDR <0.05 on the network (*Arif et al., 2021*; *Yang et al., 2021*) and performed module detection analysis using Leiden algorithm (*Traag et al., 2019*), implemented in the *leidenalg* (v0.7.0) package with 'ModularityVertexPartition'' to find the optimal partition. Modules with less than 30 genes were discarded to be able to get significant functional analysis results in the downstream analysis. To identify gene modules involved in tissue crosstalk, we correlated the module eigengenes of inter-tissue modules and assessed the significance of these correlations after Benjamini & Hochberg (BH) correction, with BH-adjusted p<0.05 was considered as significance. The module eigengene is defined as the first principal component of the expression matrix of a given module.

### Gene set functional enrichment analysis

We used Bioconductor package enrichR (v3.2) to annotate gene sets (e.g. differentially expressed genes or co-expression genes) against the curated genes sets obtained from the KEGG pathway database (*Kanehisa, 2002*), or against hallmark gene sets from MSigDB (*Liberzon et al., 2015*) using R package msigdbr (v.7.5.1). Benjamini-Hochberg (BH)–adjusted p-values (p.adj)<0.05 are considered as statistically significant and are provided in the relevant figures/datasets.

## Compass-based metabolic activity analysis

We inferred tissue-level metabolic state from RNA-seq profiles using Compass (*Wagner et al., 2021*). Compass is a flux balance analysis-based algorithm (*Orth et al., 2010*) that enables modeling the metabolic state of a system (e.g. single-cell or organ) from RNA-seq data based on a reconstructed genome-scale metabolic model of human metabolism (also known as Recon3D) (*Brunk et al., 2018*). We downloaded Recon3D from BiGG Models (http://bigg.ucsd.edu/models/Recon3D/) and created model metadata files required for the input of downstream analysis, including gene, reaction, and metabolite metadata, using in-house scripts. We used human orthologs of rat gene expression data (TPM-based) for computation of the Compass scores matrix using default parameters, and the human species as detailed in *Wagner et al., 2021*. Downstream analysis was done using the compassR R package (v.1.0.0) following the protocol (https://github.com/YosefLab/compassR; *YosefLab, 2021*).

## Statistical analyses

Data were shown as mean ± standard deviation (SD), unless stated otherwise. The assumption of normality for continuous variables from behavior tests, biometric measurements, and plasm biochemistry was determined using the Shapiro–Wilk test. Differences among multiple groups were tested by ANOVA or, for data that were not normally distributed, the non-parametric Kruskal-Wallis test. Pairwise comparisons were performed using Tukey's post hoc test following the ANOVA or Dunn's multiple comparisons post hoc test following the Kruskal-Wallis test, as appropriate. The Jaccard index was used to evaluate the similarity and diversity of two gene sets, and a hypergeometric test was used to test the significance of their overlap. All results were considered statistically significant at p<0.05, unless stated otherwise.

## Acknowledgements

The authors would like to acknowledge financial support from ScandiEdge Therapeutics and the Knut and Alice Wallenberg Foundation (No. 72110) AM and HY acknowledge support from the PoLiMeR Innovative Training Network (Marie Skłodowska-Curie Grant Agreement No. 812616) which has received funding from the European Union's Horizon 2020 research and innovation program. The computations were performed on resources provided by SNIC through the Uppsala Multidisciplinary Center for Advanced Computational Science (UPPMAX) under Project sllstore 2017024 and sctatlas.

## Additional information

### Competing interests

Matthias Uhlen, Jan Boren, Adil Mardinoglu: funder and shareholder of ScandiEdge Therapeutics that provide financial support for the study. The other authors declare that no competing interests exist.

### Funding

| Funder | Grant reference number | Author |
| --- | --- | --- |
| Knut och Alice Wallenbergs Stiftelse | 72110 | Adil Mardinoglu |
| ScandiEdge Therapeutics | | Matthias Uhlen Jan Boren Adil Mardinoglu |
| Marie Skłodowska-Curie Actions | 10.3030/812616 | Adil Mardinoglu Hong Yang |

The funders had no role in study design, data collection and interpretation, or the decision to submit the work for publication.

### Author contributions

Hong Yang, Software, Formal analysis, Investigation, Visualization, Methodology, Writing – original draft, Writing – review and editing; Cheng Zhang, Formal analysis, Supervision, Investigation, Methodology, Writing – review and editing; Woonghee Kim, Nursena Yuksel, Jihad Sebhaoui, Data curation, Investigation, Methodology, Writing – review and editing; Mengnan Shi, Formal analysis, Investigation, Methodology, Writing – review and editing; Metin Kiliclioglu, Cemil Bayram, Ismail Bolar, Özlem Özdemir Tozlu, Cem Baba, Data curation, Formal analysis, Investigation, Methodology, Writing – review and editing; Serkan Yildirim, Shazia Iqbal, Data curation, Investigation, Writing – review and editing; Ahmet Hacımuftuoglu, Resources, Data curation, Writing – review and editing; Matthias Uhlen, Jan Boren, Writing – review and editing; Hasan Turkez, Conceptualization, Resources, Data curation, Supervision, Project administration, Writing – review and editing; Adil Mardinoglu, Conceptualization, Resources, Supervision, Funding acquisition, Project administration, Writing – review and editing

### Author ORCIDs

Hong Yang ⓘ https://orcid.org/0009-0002-0414-2471
Cheng Zhang ⓘ https://orcid.org/0000-0002-3721-8586
Adil Mardinoglu ⓘ https://orcid.org/0000-0002-4254-6090

### Ethics

Oral toxicity study in rats: All the animal procedures (including housing, health monitoring, dosing, etc.) and ethical revisions were performed according to the 2010/63/EU Directive on the protection of animals used for biomedical research. JNK-IN-5A treatment study in rats: All experiments were performed in accordance with institutional ethical guidelines, and the present in vivo study was approved by the Ethics Committee for Animal Research of Atatürk University (with the reference number 2023/04 on 30.03.2023).

Reviewer #1 (Public review): https://doi.org/10.7554/eLife.98427.3.sa1
Reviewer #2 (Public review): https://doi.org/10.7554/eLife.98427.3.sa2

Author response https://doi.org/10.7554/eLife.98427.3.sa3

## Additional files

### Supplementary files

Supplementary file 1. Summary of body weight and clinical pathological parameters for 7 d of oral toxicology of JNK-IN-5A in rats.

Supplementary file 2. Summary of body weight and clinical pathological parameters for 3 wk oral administration of JNK-IN-5A in metabolic dysfunction-associated fatty liver disease (MAFLD) rats model.

Supplementary file 3. Passive avoidance data.

Supplementary file 4. Locomotor activity data.

Supplementary file 5. Elevated plus maze data.

Supplementary file 6. Result of differential expression analysis for all relevant comparisons of liver, white adipose tissue, skeletal muscle, and brain in the study, related to *Figures 2–4*.

Supplementary file 7. Gene membership in certain modules identified from the tissue-specific co-expression networks, which includes samples from control, sucrose, and sucrose+JNK_D1 groups, related to *Figure 6*.

Supplementary file 8. Results of genome-scale metabolic modeling based on Compass and Recon3D, related to *Figure 7*.

Supplementary file 9. Metabolites associated with sucrose overconsumption in MASLD.

MDAR checklist

### Data availability

All data generated or analysed during this study are included in the manuscript and supporting files. All raw RNA-sequencing data generated from this study have been deposited in GEO under accession code GSE282954.

The following dataset was generated:

| Author(s) | Year | Dataset title | Dataset URL | Database and Identifier |
|---|---|---|---|---|
| Yang H, Zhang C, Kim W, Shi M, Kiliclioglu M, Bayram C, Bolat I | 2025 | Multi-tissue network analysis reveals the effect of JNK inhibition on dietary sucrose-induced metabolic dysfunction in rats | https://www.ncbi.nlm.nih.gov/geo/query/acc.cgi?acc=GSE282954 | NCBI Gene Expression Omnibus, GSE282954 |

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
