## [Editor Report · eLife Assessment]

The authors implement a **valuable** multi-tissue approach to dissect the physiologic consequences of JNK inhibition in parallel with dietary perturbation via sucrose. The conclusions of disrupted liver, muscle and adipose metabolism being central to these effects are **solid**, as they are supported by a combination of experimental dissection and network modeling approaches.

---

## [Referee Report · Reviewer #1 (Public review)]

Summary:

In this manuscript, authors have investigated the effects of JNK inhibition on sucrose-induced metabolic dysfunction in rats. They used multi-tissue network analysis to study the effects of the JNK inhibitor JNK-IN-5A on metabolic dysfunction associated with excessive sucrose consumption. Their results show that JNK inhibition reduces triglyceride accumulation and inflammation in the liver and adipose tissues while promoting metabolic adaptations in skeletal muscle. The study provides new insights into how JNK inhibition can potentially treat metabolic dysfunction-associated fatty liver disease (MAFLD) by modulating inter-tissue communication and metabolic processes.

Strengths:

The study has several notable strengths:

Comprehensive Multi-Tissue Analysis: The research provides a thorough multi-tissue evaluation, examining the effects of JNK inhibition across key metabolically active tissues, including the liver, visceral white adipose tissue, skeletal muscle, and brain. This comprehensive approach offers valuable insights into the systemic effects of JNK inhibition and its potential in treating MAFLD.

Robust Use of Systems Biology: The study employs advanced systems biology techniques, including transcriptomic analysis and genome-scale metabolic modeling, to uncover the molecular mechanisms underlying JNK inhibition. This integrative approach strengthens the evidence supporting the role of JNK inhibitors in modulating metabolic pathways linked to MAFLD.

Potential Therapeutic Insights: By demonstrating the effects of JNK inhibition on both hepatic and extrahepatic tissues, the study offers promising therapeutic insights into how JNK inhibitors could be used to mitigate metabolic dysfunction associated with excessive sucrose consumption, a key contributor to MAFLD.

Behavioral and Metabolic Correlation: The inclusion of behavioral tests alongside metabolic assessments provides a more holistic view of the treatment's effects, allowing for a better understanding of the broader physiological implications of JNK inhibition.

Weaknesses:

The authors have adequately addressed all my concerns, and the revisions have significantly improved the manuscript's clarity and impact.

---

## [Referee Report · Reviewer #2 (Public review)]

Excessive sucrose is a possible initial factor for the development of metabolic dysfunction-associated fatty liver disease (MAFLD). To investigate the possibility that intervention with JNK inhibitor could lead to the treatment of metabolic dysfunction caused by excessive sucrose intake, the authors performed multi-organ transcriptomics analysis (liver, visceral fat (vWAT), skeletal muscle, and brain) in a rat model of MAFLD induced by sucrose overtake (+ JNK inhibitor treatment).

The major strengths and weakness of this study are as follows.

Strengths:

・It has been previously reported that inhibition of JNK signalling can contribute to the prevention of hepatic steatosis (HS) and related metabolic syndrome in other models, but the role of JNK signalling in the metabolic disruption caused by excessive intake of sucrose, a possible initial factor for the development of MAFLD, has not been well understood, and the authors have addressed this point.

・This study is also important because pharmacological therapy for MAFLD has not yet been established.

・By obtaining transcriptomic data in multiple organs and comprehensively analyzing the data using gene co-expression network (GCN) analysis and genome-scale metabolic models (GEM), the authors showed the multi-organ interaction in not only in the pathology of MAFLD caused by excessive sucrose intake but also in the treatment effects by JNK-IN-5A.

・Since JNK signalling has diverse physiological functions in many organs, the authors effectively assessed possible side effects with a view to the clinical application of JNK-IN-5A.

Weaknesses:

・The metabolic process activities were evaluated using RNA-seq results in Figure 7, but direct data such as metabolite measurements are lacking.

・There is a lack of consistency in the data between JNK-IN-5A_D1 and _D2, and there is no sufficient data-based explanation for why the effects observed in D1 were inconsistent in the D2 samples.

・Although it is valuable that the authors were able to suggest the possibility of JNK inhibitor as a therapeutic strategy for MAFLD, the evaluation of the therapeutic effect was limited to evaluation of plasma TG, LDH, and gene expression changes. As there was no evaluation of liver tissue images, it is unclear what changes were brought about in the liver by the excessive sucrose intake and the treatment with JNK-IN-5A.

As mentioned in the Weakness section, biological data is insufficient, such as the lack of metabolite measurements and a histological evaluation of the liver. However, overall, the authors successfully provided the valuable insights that the JNK inhibitor has a cross-organ therapeutic effect on their MAFLD model induced by sucrose overtake. Their insist is supported by convincing data, comprehensively analysing the transcriptomic data obtained from multiple organs using GCN (gene co-expression network) analysis and GEM (genome-scale metabolic modelling).

Their comprehensive transcriptomic analysis in multiple organs, including the brain, has demonstrated that the effects of drugs are more widespread than just on specific tissues thought to be the main target, indicating the importance of focusing on tissue interactions when we assess the effects of drugs. Also, the data set in this study will be useful for comparative evaluation with transcriptomics data for other MALFD models.

---

## [Author Response]

The following is the authors’ response to the original reviews.

**Public Reviews:**

**Reviewer #1 (Public review):**
Summary:In this manuscript, authors have investigated the effects of JNK inhibition on sucrose-induced metabolic dysfunction in rats. They used multi-tissue network analysis to study the effects of the JNK inhibitor JNK-IN-5A on metabolic dysfunction associated with excessive sucrose consumption. Their results show that JNK inhibition reduces triglyceride accumulation and inflammation in the liver and adipose tissues while promoting metabolic adaptations in skeletal muscle. The study provides new insights into how JNK inhibition can potentially treat metabolic dysfunction-associated fatty liver disease (MAFLD) by modulating inter-tissue communication and metabolic processes.Strengths:The study has several notable strengths:Comprehensive Multi-Tissue Analysis: The research provides a thorough multi-tissue evaluation, examining the effects of JNK inhibition across key metabolically active tissues, including the liver, visceral white adipose tissue, skeletal muscle, and brain. This comprehensive approach offers valuable insights into the systemic effects of JNK inhibition and its potential in treating MAFLD.Robust Use of Systems Biology: The study employs advanced systems biology techniques, including transcriptomic analysis and genome-scale metabolic modeling, to uncover the molecular mechanisms underlying JNK inhibition. This integrative approach strengthens the evidence supporting the role of JNK inhibitors in modulating metabolic pathways linked to MAFLD.Potential Therapeutic Insights: By demonstrating the effects of JNK inhibition on both hepatic and extrahepatic tissues, the study offers promising therapeutic insights into how JNK inhibitors could be used to mitigate metabolic dysfunction associated with excessive sucrose Behavioral and Metabolic Correlation: The inclusion of behavioral tests alongside metabolic assessments provides a more holistic view of the treatment's effects, allowing for a better understanding of the broader physiological implications of JNK inhibition.Weaknesses:While the study provides a comprehensive evaluation of JNK inhibitors in mitigating MAFLD conditions, addressing the following points will enhance the manuscript's quality:The authors should explicitly mention and provide a detailed list of metabolites affected by sucrose and JNK inhibition treatment that have been previously associated with MAFLD conditions. This will better contextualize the findings within the broader field of metabolic disease research.

We fully agreed on this constructive suggestion to improve our understanding of the metabolic effect of JNK inhibition under sucrose overconsumption. While technical limitations made it challenging to directly analyze metabolites in the current study, we employed genome-scale metabolic modeling—a robust approach for studying metabolism—to predict the metabolic pathways potentially impacted by the interventions (Fig. 7 and Data S8). Additionally, as part of this revision, we conducted an extensive literature review to identify metabolites previously reported to be affected by sucrose consumption in MAFLD rodent models and MASLD patients. A detailed summary of these metabolites is now presented in attached Table 1 and several of these metabolites have been incorporated into the revised results section (Lines 308-314) to support some of the predicted metabolic activities.

“Some of the predicted metabolic changes align with previous findings in rodents subjected to sucrose overconsumption. For example, Öztürk et al. reported altered tryptophan metabolism, including decreased serum levels of kynurenic acid and kynurenine, in rats consuming 10% sucrose in drinking water. Similarly, increased triglyceride-bound oleate, palmitate, and stearate were observed in the livers of rats fed a 10% sucrose solution, indicating JNK-IN-5A treatment may regulate lipid metabolism by modulating these metabolic activities.”

It is important to note, however, that data on metabolites specifically affected by JNK inhibition in MASLD contexts remains lacking in the literature. The predicted metabolites and associated metabolic pathways in the current study could provide a starting point for such exploration in future studies. We have emphasized this in the revised manuscript and highlighted the need for further studies to explore these mechanisms in greater detail.

**Author response table 1. sa3table1:** Metabolites associated with sucrose overconsumption in MASLD.

Study	Intervention(s) and duration	Metabolites	Metabolism
Fujii et al., 2024 [1].	Male rats fed a 10% sucrose solution for 16 days	Decreased blood acetate and butyrate levels	Gut microbial metabolism, Lipid metabolism
Sun et al., 2021 [2].	Male Wistar rats fed high-sucrose diet for 4 weeks	Decreased butyrate and formate in the cecal content	
Song et al., 2024 [3].	Male Wistar rats fed a high sucrose diet for 4 weeks	Decreased acetate and butyrate; increased succinate in cecal content	
Ramos-Romero et al., 2019 [4].	Male Wistar rats were fed a 35% sucrose solution for 24 weeks.	Increased uric acid in urine	Urea cycle
Field, 2023[5].	Male C57BL/6J mice were fed either low-fat diet or high-fat diet with 30% sucrose water for 8 weeks	Elevated serum and kidney erythritol concentration	Carbohydrate metabolism
Beckmann et al. 2015 [6].	Healthy females (n=90) consumed 0,50 , or 100 g sucrose in water, followed by urine and blood sampling at 0,3 , and 24 hours.	Increased erythronic acid in plasma	Carbohydrate metabolism
He et al., 2023 [7].	Male C57BL/6J mice were fed a high sucrose diet (30% sucrose in drinking water) for 24 weeks	Decreased muricholic acid level in the liver, cecal and colon content. increased hyocholic acid levels in the serum.	Bile acid metabolism
Stephenson et al. 2022 [8].	Mice fed 10% sucrose solution in drinking water for 12 weeks	Increased triglyceridebound oleate, palmitate, and stearate in liver; Mixed alteration in serum bile acids pool (sex and treatment interaction effect)	Lipid metabolism and bile acid metabolism
Mock et al. 2017 [9].	Female rats fed 13% sucrose solutions for 8 weeks	Increased palmitoleic acid in gonadal and retroperitoneal fat pads; higher serum triglyceride	Lipid metabolism
Oztūrk et al. 2022 [10].	Wistar male rats fed 10% sucrose in drinking for 3 months	Decreased serum levels of kynurenic acid and kynurenine	Tryptophan metabolism
Gariani, Karim, et al. 2016 [11].	Male C57BL/6J mice were fed with a Western high-fat and highsucrose	Decreased NAD* levels in the liver	Fatty acid oxidation
Togo et al., 2019 [12].	C57BL/6J mice fed liquid (50% by weigh) or solid sucrose for 8 weeks	Elevated hepatic fat	Lipid metabolism

The limitations of the study should be clearly stated, particularly the lack of evidence on the effects of chronic JNK inhibitor treatment and potential off-target effects. Addressing these concerns will offer a more balanced perspective on the therapeutic potential of JNK inhibition.

Thank you for this constructive comment. We have acknowledged limitations of the current study in Discussion section (Lines 397-406) of the revised manuscript:

“Nevertheless, several limitations warrant consideration. First, while we observed transcriptional adaptations in skeletal muscle tissue following treatment, the exact molecular mechanisms underlying these changes and their roles in skeletal muscle function and systemic metabolic homeostasis remain unclear. Further investigation is warranted to elucidate the muscle-specific effects of JNK inhibition. Second, our study did not investigate the dosedependent or potential off-target effects of JNK-IN-5A, particularly its activity on other members of the kinase family and associated signaling pathways. Lastly, the long-term effects of JNKIN-5A administration remain unexplored. Understanding its prolonged impact across different stages of MAFLD, including advanced MASH, is crucial for assessing the full therapeutic potential of JNK inhibition in the treatment of MAFLD.“

The potential risks of using JNK inhibitors in non-MAFLD conditions should be highlighted, with a clear distinction made between the preventive and curative effects of these therapies in mitigating MAFLD conditions. This will ensure the therapeutic implications are properly framed.

Thank you for this insightful suggestion. The potential risks of using JNK inhibitors in nonMAFLD conditions have been considered and are now highlighted in Lines 369-390 of the revised discussion

“Although overactivated JNK activity presents an attractive opportunity to combat MAFLD, inhibition of JNK presents substantial challenges and potential risks due to its broad and multifaceted roles in many cellular processes. One key challenge is the dual role of JNK signaling (Lamb et al., 2003). For instance, long-term JNK inhibition may disrupt liver regeneration, as JNK plays a critical role in liver repair by regulating hepatocyte proliferation and survival following injury or stress (Papa and Bubici, 2018). In HCC, it has been reported that JNK acts as both a tumor promoter, driving inflammation, fibrosis, and metabolic dysregulation, and a tumor suppressor, facilitating apoptosis and cell cycle arrest in damaged hepatocytes. Its inhibition, therefore, carries the risk of inadvertently promoting tumor progression under certain conditions (Seki et al., 2012). Furthermore, the differential roles of JNK isoforms (JNK1, JNK2, JNK3) and a lack of specificity of JNK inhibitors present another layer of complexity. Given these challenges, while our study demonstrated the potential of JNK-IN-5A in mitigating early metabolic dysfunction in the liver and adipose tissues, JNK targeting strategies should be carefully tailored to the disease stage under investigation. For curative approaches targeting advanced MAFLD, such as MASH, future studies are warranted to address considerations related to dosing, tissue specificity, and the long-term effects.”

The statistical analysis section could be strengthened by providing a justification for the chosen statistical tests and discussing the study's power. Additionally, a more detailed breakdown of the behavioral test results and their implications would be beneficial for the overall conclusions of the study.

We would like to thank you for this constructive suggestion. In this study, differences among more than two groups were tested using ANOVA or Kruskal-Wallis test based on the normality testing (Shapiro–Wilk test) on the data (continuous variables from different measurements). Pairwise comparisons, were performed using Tukey’s post hoc test following ANOVA or Dunn’s multiple comparisons post hoc test following the Kruskal-Wallis test, as appropriate.

The study used 11 animals per group, a group size widely used in preclinical animal research [13]. To evaluate the power of this study design to detect group differences, we conducted a power analysis using G*Power 3.1 software [14], with ANOVA used as an example. The power analysis revealed the following:

- For a small effect size (partial eta.sq = 0.01), the power was 7.5% at 𝑝<0.05.

- For a medium effect size (partial eta.sq = 0.06), the power was 23.7% at 𝑝<0.05.

- For a large effect size (partial eta.sq = 0.14), the power is 55.4% at 𝑝<0.05

Bonapersona et al. reported that the median statistical power in animal studies is often between 15–22% [15], the achieved power of the current study design is within the range observed in most exploratory animal research. However, we acknowledge that the power for detecting smaller effects within groups is limited, which is also a common challenge in animal research due to ethical considerations on increasing sample sizes.

As suggested, we’ve revised the ‘Statistical Analysis’ and ‘Result’ sections to improve clarity:

“Statistical Analysis:

Data were shown as mean ± standard deviation (SD), unless stated otherwise. The assumption of normality for continuous variables from behavior test, biometric measurements, and plasm biochemistry was determined using the Shapiro–Wilk test. Differences among multiple groups were tested by ANOVA or, for data that were not normally distributed, the non-parametric Kruskal-Wallis test. Pairwise comparisons were performed using Tukey’s post hoc test following the ANOVA or Dunn’s multiple comparisons post hoc test following the Kruskal-Wallis test, as appropriate. The Jaccard index was used to evaluate the similarity and diversity of two gene sets, and a hypergeometric test was used to test the significance of their overlap. All results were considered statistically significant at p < 0.05, unless stated otherwise.”

Behavior tests (Lines 150-157):

“We found no significant differences among groups in retention latencies, a measure of learning and memory abilities in passive avoidance test (Data S3). Additionally, the locomotor activity test was used to analyze behaviors such as locomotion, anxiety, and depression in rat. No significant differences were observed among groups in stereotypical movements, ambulatory activity, rearing, resting percentage, and distance travelled (Data S4). Similarly, the elevated plus maze test (Walf and Frye, 2007), an assay for assessing anxiety-like behavior in rodents, showed that rats in all groups had comparable open-arm entries and durations (Data S5). Collectively, the behavior tests indicate the JNK-IN-5A-treated rats exhibit no evidence of anxiety and behavior disorders.”

**Reviewer #2 (Public review):**
Summary:Excessive sucrose is a possible initial factor for the development of metabolic dysfunctionassociated fatty liver disease (MAFLD). To investigate the possibility that intervention with JNK inhibitor could lead to the treatment of metabolic dysfunction caused by excessive sucrose intake, the authors performed multi-organ transcriptomics analysis (liver, visceral fat (vWAT), skeletal muscle, and brain) in a rat model of MAFLD induced by sucrose overtake (+ a selective JNK2 and JNK3 inhibitor (JNK-IN-5A) treatment). Their data suggested that changes in gene expression in the vWAT as well as in the liver contribute to the pathogenesis of their MAFLD model and revealed that the JNK inhibitor has a cross-organ therapeutic effect on it.Strengths:(1)It has been previously reported that inhibition of JNK signaling can contribute to the prevention of hepatic steatosis (HS) and related metabolic syndrome in other models, but the role of JNK signaling in the metabolic disruption caused by excessive intake of sucrose, a possible initial factor for the development of MAFLD, has not been well understood, and the authors have addressed this point.(2)This study is also important because pharmacological therapy for MAFLD has not yet been established.(3)By obtaining transcriptomic data in multiple organs and comprehensively analyzing the data using gene co-expression network (GCN) analysis and genome-scale metabolic models (GEM), the authors showed the multi-organ interaction in not only in the pathology of MAFLD caused by excessive sucrose intake but also in the treatment effects by JNK-IN-5A.(4) Since JNK signaling has diverse physiological functions in many organs, the authors effectively assessed possible side effects with a view to the clinical application of JNK-IN-5A.Weaknesses:(1) The metabolic process activities were evaluated using RNA-seq results in Figure 7, but direct data such as metabolite measurements are lacking.

Thank you for these valuable insights. We fully agree that direct metabolite measurements would provide a deeper understanding of the metabolic impact of sucrose overconsumption and JNK-IN-5A administration. Unfortunately, due to technical limitations, we were unable to directly measure metabolites in this study. To address this, we supported our genome-scale metabolic modeling predictions with an extensive literature review, which is summarized in attached Table 1. This table highlights key metabolites and associated metabolic pathways that have been previously associated with sucrose overconsumption in MAFLD contexts. We incorporated some of these metabolites into the revised results section (Lines 308–314) to demonstrate the consistency between our predicted metabolic changes and experimental findings from the literature. For instance, studies have reported altered tryptophan metabolism, including decreased serum kynurenic acid and kynurenine levels, as well as increased triglyceride-bound oleate, palmitate, and stearate in sucrose-fed rodents. These findings align with our predictions of altered metabolic activities in fatty acid oxidation, fatty acid synthesis, and tryptophan metabolism.

(2) There is a lack of consistency in the data between JNK-IN-5A_D1 and _D2, and there is no sufficient data-based explanation for why the effects observed in D1 were inconsistent in the D2 samples.

Thank you for raising this important point regarding the differences between the two dosages. As this was not the primary focus of the current study and we do not have sufficient data to fully explain these observations. Our speculation is that this may arise from pharmacokinetic differences associated with the dosing of this small molecule inhibitor, including potential saturation of transport mechanisms, alter tissue distribution, or off-target effects.

(3) Although it is valuable that the authors were able to suggest the possibility of JNK inhibitor as a therapeutic strategy for MAFLD, the evaluation of the therapeutic effect was limited to the evaluation of plasma TG, LDH, and gene expression changes. As there was no evaluation of liver tissue images, it is unclear what changes were brought about in the liver by the excessive sucrose intake and the treatment with JNK-IN-5A.

We acknowledge that the lack of histological evaluations may limit to having a complete picture of the interventions' effects. However, as you noted, our transcriptional and systems-wide investigation across multiple tissues provides novel and significant insights into the molecular and systemic impacts of JNK-IN-5A treatment.

**Recommendations for the authors:**

**Reviewer #2 (Recommendations for the authors):**
(1) It would be useful to explain why the authors conducted their research using female rats but not male rats.

Thank you for raising this insightful point. We chose female rats for the current study was based on several considerations. (1) Previous research has demonstrated that female rats exhibit metabolic dysfunction (e.g., hypertriglyceridemia, liver steatosis, insulin resistance) in response to dietary factors, such as high-sucrose feeding [16-19]. These metabolic characteristics made them an appropriate model for assessing the in vivo effects of JNK inhibition under high-sucrose conditions. (2) It is also reported that female rats show resilience to high-sucrose-induced metabolic dysfunction due to the protective effects of estrogen [8], we aimed to determine whether JNK inhibition could provide therapeutic benefits in this context. This allows us to evaluate the effect of JNK inhibition even in metabolically advantaged groups. (3) Our results from the tolerance test (Fig. 2a) indicated that female rats displayed more fluctuating variation to JNK-IN-5A administration. This variation allowed us to evaluate how JNK inhibition influences metabolic outcomes in a sex that is more responsive to the intervention. Nonetheless, we emphasize the importance of future studies involving male rats to better understand sex-specific responses to JNK inhibition and to provide more comprehensive guidance for the development of JNK-targeting therapies in MAFLD treatment.

(2) Figure 2C shows that JNK-IN-5A administration reduces the mRNA levels of Mapk8 and Mapk9 in the liver and the SkM. It would be useful to provide the authors' insight into the data.

In the liver, the data in Figure 2 in original submission and the attached Fig. 1 show that sucrose feeding induces opposite alterations in the mRNA expression of *Mapk8* (*Jnk1*, increased, log2FC_SucrosevsControl_ = 0.02) and *Mapk9* (*Jnk2*, decreased, log2FC_SucrosevsControl_ = -0.43), though these changes do not reach statistical significance. JNK-IN-5A administration reverses these effects, significantly decreasing *Mapk8* expression (log2FC_Sucrose+JNK_D1vsSucrose_ = -0.37) while increasing *Mapk9* expression (log2FC_Sucrose+JNK_D1vsSucrose_ = 0.42). This suggests potential differential yet compensatory roles of these two isoforms in regulating JNK activity during these interventions in the liver, keeping in line with the findings from *Jnk1*- and/or *Jnk2*-specific knockout studies [20, 21]. Additionally, emerging evidence indicates that *Jnk1* plays a major role in diet-induced liver fibrosis and metabolic dysfunction [22-25]. Therefore, the reduced *Mapk8* expression following JNK-IN-5A administration may contribute to the observed improvements in liver metabolism.

**Author response image 1. sa3fig1:** The spearman correlation between expression levels of *Mapk8.*

In skeletal muscle, the primary site for insulin-stimulated glucose uptake, insulin signaling is crucial for maintaining metabolic homeostasis [26]. Numerous studies have demonstrated that JNK activation promotes insulin resistance and targeting JNK might be a promising therapeutic strategy for the treatment of metabolic diseases associated with insulin resistance, such as MAFLD [24]. In our study, while sucrose overconsumption did not significantly alter the mRNA levels of JNK isoforms in this tissue, JNK-IN-5A at dosage 30 mg/kg/day administration significantly reduced the expression of both *Jnk1* and *Jnk2* as well as genes involved in insulin signaling (Fig. 5). This suggests a potential interplay between JNK inhibition and insulin signaling pathways in the skeletal muscle, where inhibition of JNK activity may improve insulin sensitivity by modulating these pathways. However, it is also crucial to investigate the longterm effects of JNK-IN-5A administration and its broader impact on many other physiological processes regulated by the JNK pathway. These aspects will be a focus of our future studies.

(3) The notations a and b in Figure S5 are missing.

Thank you for this constructive comment. We have corrected this in the revised figure S5.

(4) Data S13 described in the figure legend for Figure 7 (lines 630 and 632) seems a mistake and should be Data S8.(5) The notations a, b, and c in Figure 7 are incorrect. The figure legend for Figure 7a doesn't seem to match the figure contents.

We appreciate your attention to details regarding Fig. 7. We have corrected the reference and the figure legend in revised Fig. 7.

Reference

(1) Fujii, A., et al., Sucrose Solution Ingestion Exacerbates DinitrofluorobenzeneInduced Allergic Contact Dermatitis in Rats. Nutrients, 2024. 16(12).(2) Sun, S., et al., High sucrose diet-induced dysbiosis of gut microbiota promotes fatty liver and hyperlipidemia in rats*.* J Nutr Biochem, 2021. 93: p. 108621.(3) Qi, S., et al., Inositol and taurine ameliorate abnormal liver lipid metabolism induced by high sucrose intake*.* Food Bioscience, 2024. 60: p. 104368.(4) Ramos-Romero, S., et al., The Buckwheat Iminosugar d-Fagomine Attenuates Sucrose-Induced Steatosis and Hypertension in Rats*.* Mol Nutr Food Res, 2020. 64(1): p. e1900564.(5) Ortiz, S.R. and M.S. Field, Sucrose Intake Elevates Erythritol in Plasma and Urine in Male Mice*.* J Nutr, 2023. 153(7): p. 1889-1902.(6) Beckmann, M., et al., Changes in the human plasma and urinary metabolome associated with acute dietary exposure to sucrose and the identification of potential biomarkers of sucrose intake*.* Mol Nutr Food Res, 2016. 60(2): p. 444-57.(7) He, X., et al., High Fat Diet and High Sucrose Intake Divergently Induce Dysregulation of Glucose Homeostasis through Distinct Gut Microbiota-Derived Bile Acid Metabolism in Mice*.* J Agric Food Chem, 2024. 72(1): p. 230-244.(8) Stephenson, E.J., et al., Chronic intake of high dietary sucrose induces sexually dimorphic metabolic adaptations in mouse liver and adipose tissue*.* Nat Commun, 2022. 13(1): p. 6062.(9) Mock, K., et al., High-fructose corn syrup-55 consumption alters hepatic lipid metabolism and promotes triglyceride accumulation. J Nutr Biochem, 2017. 39: p. 32-39.(10) Eryavuz Onmaz, D. and B. Ozturk, Altered Kynurenine Pathway Metabolism in Rats Fed Added Sugars*.* Genel Tıp Dergisi, 2022. 32(5): p. 525-529.(11) Gariani, K., et al., Eliciting the mitochondrial unfolded protein response by nicotinamide adenine dinucleotide repletion reverses fatty liver disease in mice*.* Hepatology, 2016. 63(4): p. 1190-204.(12) Togo, J., et al., Impact of dietary sucrose on adiposity and glucose homeostasis in C57BL/6J mice depends on mode of ingestion: liquid or solid*.* Mol Metab, 2019. 27: p. 22-32.(13) Arifin, W.N. and W.M. Zahiruddin, Sample Size Calculation in Animal Studies Using Resource Equation Approach*.* Malays J Med Sci, 2017. 24(5): p. 101-105.(14) Faul, F., et al., G*Power 3: a flexible statistical power analysis program for the social, behavioral, and biomedical sciences*.* Behav Res Methods, 2007. 39(2): p. 175-91.(15) Bonapersona, V., et al., Increasing the statistical power of animal experiments with historical control data*.* Nat Neurosci, 2021. 24(4): p. 470-477.(16) Kendig, M.D., et al., Metabolic EYects of Access to Sucrose Drink in Female Rats and Transmission of Some EYects to Their OYspring*.* PLoS One, 2015. 10(7): p. e0131107.(17) Harris, R.B.S., Source of dietary sucrose influences development of leptin resistance in male and female rats*.* Am J Physiol Regul Integr Comp Physiol, 2018. 314(4): p. R598-R610.(18) Velasco, M., et al., Sexual dimorphism in insulin resistance in a metabolic syndrome rat model*.* Endocr Connect, 2020. 9(9): p. 890-902.(19) Maniam, J., C.P. Antoniadis, and M.J. Morris, The eYect of early-life stress and chronic high-sucrose diet on metabolic outcomes in female rats*.* Stress, 2015. 18(5): p. 524-37.(20) Singh, R., et al., DiYerential eYects of JNK1 and JNK2 inhibition on murine steatohepatitis and insulin resistance*.* Hepatology, 2009. 49(1): p. 87-96.(21) Sabapathy, K., et al., Distinct roles for JNK1 and JNK2 in regulating JNK activity and c-Jun-dependent cell proliferation*.* Mol Cell, 2004. 15(5): p. 713-25.(22) Zhao, G., et al., Jnk1 in murine hepatic stellate cells is a crucial mediator of liver fibrogenesis*.* Gut, 2014. 63(7): p. 1159-72.(23) Czaja, M.J., JNK regulation of hepatic manifestations of the metabolic syndrome*.* Trends Endocrinol Metab, 2010. 21(12): p. 707-13.(24) Solinas, G. and B. Becattini, JNK at the crossroad of obesity, insulin resistance, and cell stress response*.* Mol Metab, 2017. 6(2): p. 174-184.(25) Schattenberg, J.M., et al., JNK1 but not JNK2 promotes the development of steatohepatitis in mice*.* Hepatology, 2006. 43(1): p. 163-72.(26) Sylow, L., et al., The many actions of insulin in skeletal muscle, the paramount tissue determining glycemia*.* Cell Metab, 2021. 33(4): p. 758-780.